# Causal Inference in the Closed-Loop: Marginal Structural Models for Sequential Excursion Effects

**Alexander W. Levis**[*]
Carnegie Mellon University
alevis@cmu.edu

**Gabriel Loewinger**[*]
National Institutes of Health
gloewinger@gmail.com

**Francisco Pereira**
National Institutes of Health
francisco.pereira@nih.gov

## Abstract

Optogenetics is widely used to study the effects of neural circuit manipulation on behavior. However, the paucity of causal inference methodological work on this topic has resulted in analysis conventions that discard information, and constrain the scientific questions that can be posed. To fill this gap, we introduce a nonparametric causal inference framework for analyzing "closed-loop" designs, which use dynamic policies that assign treatment based on covariates. In this setting, standard methods can introduce bias and occlude causal effects. Building on the sequentially randomized experiments literature in causal inference, our approach extends history-restricted marginal structural models for dynamic regimes. In practice, our framework can identify a wide range of causal effects of optogenetics on trial-by-trial behavior, such as, fast/slow-acting, dose-response, additive/antagonistic, and floor/ceiling. Importantly, it does so without requiring negative controls, and can estimate how causal effect magnitudes evolve across time points. From another view, our work extends "excursion effect" methods—popular in the mobile health literature—to enable estimation of causal contrasts for treatment sequences greater than length one, in the presence of positivity violations. We derive rigorous statistical guarantees, enabling hypothesis testing of these causal effects. We demonstrate our approach on data from a recent study of dopaminergic activity on learning, and show how our method reveals relevant effects obscured in standard analyses.

## 1 Introduction

Optogenetics is a neuroscience technique to "turn on/off" neurons *in vivo* in real-time, with millisecond time resolution. It works by shining lasers on neurons that have been genetically modified through viral infection to express a light-sensitive protein. It is one of the most popular assays with roughly 700 references to it in 2023 alone.[2] Optogenetics is often applied to study the causal effect of manipulating specific brain circuits while animals (e.g., mice) perform behavioral tasks to study, for example, learning and decision-making. These tasks are typically composed of a sequence of trials, $t \in \{1, 2, ..., T\}$, each of which involves presentation of stimuli and an opportunity for a behavioral response. For example, a trial might begin with a cue (e.g., a light), which indicates that a lever press will trigger delivery of a food reward. Investigators might want to know, for instance, whether applying optogenetic stimulation on a random subset of trials alters the rate at which mice press the lever. On trial $t$, an animal's behavioral outcome, $Y_t$, time-varying covariates, $X_t$, and optogenetic (treatment) indicator, $A_t$, are observed. Experiments often include both treatment ($G = 1$) and negative-control ($G = 0$) groups, with animals assigned randomly to each. While the laser (i.e., the sequential treatment, $A_t$) is often applied on a random[3] subset of trials in

---

[*]Authors contributed equally, names alphabetized

[2]699 papers on Web of Science mention "optogenetics" in 2023 and between 526-796 in years 2015-2023.

[3]Some studies deterministically set laser/no-laser trials but we focus on "stochastic" experimental designs.

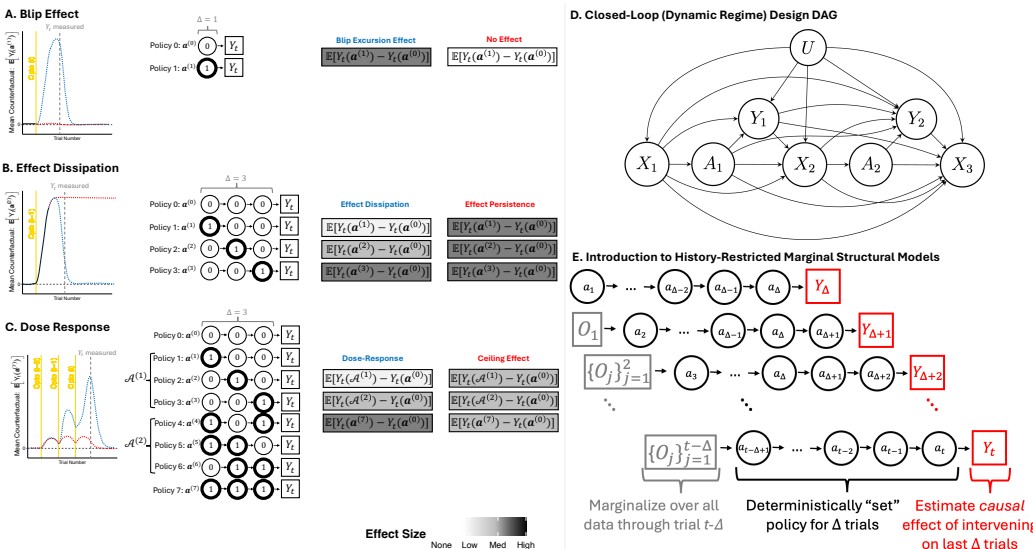

Figure 1: *Sequential Excursion Effects*. [A]-[C] The left panels show one setting where a sequence of laser simulations do or do not have the indicated effect on the outcome. The middle panel shows deterministic static policies that could be used to construct a causal contrast to probe the effect. The right panel shows what the anticipated effect size (darker is larger) of the contrast might be if the effect was or was not present. [A] **Blip Effect**: the effect of a single stimulation vs. no treatment on a recent trial. [B] **Effect Dissipation**: Whether the effect of a single stimulation causes an effect that rises and dissipates after a few trials, or persists. [C] **Dose Response**: Do successive simulations increase the response in a dose-dependent fashion? [D] Closed-loop design DAG for two trials. $U$ is an unmeasured variable. [E] HR-MSM illustration inspired by figure in [6].

both groups, only treatment group animals ($G = 1$) express the protein that enables the laser to trigger the target neural response. The control group thus controls for "off-target" effects such as the laser heating the brain, and the optogenetic insertion surgery. To answer the question above, investigators often estimate the effect of optogenetic manipulation through comparisons such as $\psi_t = \mathbb{E}[Y_t \mid G = 1] - \mathbb{E}[Y_t \mid G = 0]$. It is common to test whether $\psi_t = 0$ at specific timepoints like the end of the study ($t = T$), or to conduct inference on summaries (e.g., $\bar{\psi} = \frac{1}{T} \sum_t \psi_t$). These *between-group* comparisons assess the intervention impact based on simple long-term, or "macro"/"global" longitudinal effects. When studies randomly deliver treatment at each trial (i.e., with stochastic policies), *within-group* comparisons between laser and no-laser trials are also common (e.g., $\tilde{\psi} = \sum_t \{\mathbb{E}[Y_t \mid A_t = 1, \ G = 1] - \mathbb{E}[Y_t \mid A_t = 0, \ G = 1]\}$).

Importantly, such comparisons do not lend themselves to testing *within-group* "micro"/"local" longitudinal effects related to specific treatment sequence patterns. For example, one might ask whether there is a dose-dependent relationship between the outcome and the number of stimulations in the last five trials, or whether stimulation on two consecutive trials has a synergistic effect that is greater than if stimulation instead occurred on two non-consecutive trials. Figures 1A-C shows some representative micro longitudinal effects that are identifiable in many optogenetics studies, yet typically are not explored. Critically, such effects may be present even in studies in which one fails to detect the macro effects commonly tested. However, no formal causal inference framework has been applied to these studies, resulting in analysis conventions that limit the scope of questions researchers can ask.

Furthermore, certain experimental designs can complicate the use and interpretation of even standard analysis approaches. In "closed-loop" (referred to as "dynamic regimes" in the causal inference literature) designs, stimulation is applied depending on the behavior of the animal. For example, say a study tests if lever pressing for food, $Y_t$, decreases if optogenetic stimulation ($A_t = 1$) is applied, with positive probability, only when animals approach the lever ($X_t = 1$). Since $A_t$ is randomized conditional on $X_t$, one must incorporate $X_t$ into their analysis, but standard strategies like including $X_t$ as a covariate in a regression can obscure effects and induce bias. This is because 1) $X_t$ influences the probability of *both* the outcome and treatment, and thus can be cast as a time-varying confounder, and 2) $X_t$ *also* mediates the effect of prior treatments [7]; see the illustrative DAG in Figure 1D, though note that we generalize this setting later on to allow treatment to depend on the

*complete* history of previously measured variables at each time point. In this case, since treatment also influences both $Y_t$ and $X_t$ on subsequent trials, closed-loop designs induce "treatment–confounder feedback" [7], which can lead to bias with standard analyses. We include an example in Appendix B to show how, when the treatment has opposing effects on $Y_t$ and $X_t$, treatment and control groups can exhibit *identical* average (observed) outcome levels even if the laser causes a large immediate effect. Furthermore, standard regression approaches can actually induce collider-bias, and block mediators of the treatment effect [7]. Finally, if treatment policies deterministically rule out treatment (e.g., when $X_t = 0$), certain effects are not identifiable: the positivity violation inherent to these designs precludes estimation of certain counterfactual distributions. Closed-loop designs therefore require specialized algorithms for valid causal inference.

More broadly, there have been a number of high profile calls for more rigorous definitions of causality and causal inference in neuroscience [1, 33, 4, 16]. However, to the best of our knowledge, existing methodological work [25, 10, 14] focuses on instrumental variable-based approaches to estimate causal effects of optogenetics on neural activity. Unlike our setting, these methods are restricted to datasets that include both measurements of the activity of the neurons stimulated by optogenetics, and the neurons those cells interact with. One can then conceptualize the neural activity of the stimulated neurons as treatment variables, and the optogenetics sequence as instruments. In addition to focusing on behavioral outcomes, we explicitly deal with sequentially randomized (and closed-loop) designs, whereas prior work treats each trial as an exchangeable draw, ignoring the sequential nature of trials.

Our contributions are (1) proposing the first formal counterfactual-based causal framing of these behavioral optogenetics designs, (2) developing an analysis framework based on history-restricted marginal structural models that enables the estimation of "sequential excursion effects" that capture the local causal contrasts described above, (3) expanding excursion effect methodology to account for positivity violations, and to accommodate treatment sequences greater than length one, (4) providing estimators with efficient computational implementations and strong theoretical guarantees under minimal nonparametric conditions (verified in simulations), and (5) applying our methods to data from a high profile *Nature* paper, and showing how they reveal effects obscured by standard methods.

## 2 Notation and Related Work

In this section, we (i) provide the necessary notation and a brief review of relevant work, and (ii) describe the key methodological gap in the current literature: existing methods cannot estimate causal effects of proximal treatment sequences longer than one timepoint in closed-loop designs.

**Notation**   Let $\mathcal{O}_t = \{X_t, A_t, Y_t\}$ be the vector of *observed* variables for an animal on trial $t$. We denote $T$ as the number of trials and $[T]$ as the set $\{1, 2, ..., T\}$. A sample of subjects $i = 1, 2, ..., n$ is collected but, as subjects are exchangeable, we often suppress indices to reduce notational burden. We express *counterfactual* variables, or *potential outcomes*, with parentheses. For example, $Y_t(\boldsymbol{a}_t)$, represents the potential outcome that would be observed at trial $t$ if a subject received the treatment sequence, $\boldsymbol{a}_t = (a_1, \ldots, a_t)$. Overbars represent all history up to and including a given trial. For example, $\overline{B}_j = (B_1, \ldots, B_j)$, for any sequence of variables $\{B_t\}_{t=1}^T$, and any $j \in [T]$. Finally, we define $H_t = (\overline{X}_t, \overline{A}_{t-1}, \overline{Y}_{t-1})$, so $H_t$ includes all information prior to the treatment "decision" at $t$.

**Relevant Literature**   *Marginal structural models* (MSMs) are often used to model the mean counterfactuals $\mathbb{E}[Y_t(\boldsymbol{a}_t)]$[28, 30, 29] in sequentially randomized experiments, though these typically do not perform well [21] when there are a large number of time points (e.g., as in many optogenetics studies): the variance of the model coefficients can grow prohibitively large. *History-restricted* MSMs [21] (HR-MSMs) model $\mathbb{E}[Y_t(\boldsymbol{a}_{\Delta,t})]$ for some $\boldsymbol{a}_{\Delta,t} = (a_{t-\Delta+1}, \ldots, a_t)$, typically with $\Delta \ll t$. That is, HR-MSMs model the mean counterfactual outcome at time $t$, under an intervention defined on a proximal (often short) treatment sequence. As $\mathbb{E}[Y_t(\boldsymbol{a}_{\Delta,t})] = \mathbb{E}[Y_t(\overline{A}_{t-\Delta}, \boldsymbol{a}_{\Delta,t})]$, by a consistency assumption, these estimands implicitly marginalize over the observed treatment sequence, $\overline{A}_{t-\Delta}$, prior to the first point of intervention. However, any Markov-like assumptions made by the causal framework follow directly from the experimental design: HR-MSMs (and, by extension, our proposed methods) allow for $X_t$, $Y_t$ to be causally affected by *all* prior trials (i.e., $\mathcal{O}_j$ for $j \in [t-1]$). By placing structure on $\mathbb{E}[Y_t(\boldsymbol{a}_{\Delta,t})]$, the HR-MSM can borrow strength across treatment sequences $\boldsymbol{a}_{\Delta,t}$, which can increase power when there are many trials. Figure 1E provides a graphical illustration of HR-MSMs. HR-MSMs are typically fit by using generalized estimating equations (GEE) with

inverse probability of treatment weighting (IPW). IPW resolves the dilemma with standard regression techniques in sequentially randomized experiments, outlined above, where failure to condition on time-varying confounders, $X_t$, biases estimates (as treatment is randomized conditional on $X_t$ in closed-loop designs), but conditioning on $X_t$ induces confounding ($X_t$ are colliders on the path between past treatments and subsequent outcomes, through unmeasured confounders, $U$, as shown in the DAG in Figure 1D)) [7]. HR-MSMs can also incorporate time-varying effect modifiers (e.g., see [23] and references), to test, for example, whether causal effects vary across trials, or animal-specific covariate levels.

**The gaps: Sequential effects and positivity violations**  In designs that assign treatment randomly at each trial, HR-MSMs can be used to estimate the causal effect of specific deterministic treatment sequences $\boldsymbol{a}_{\Delta,t}$ that may differ from the observed sequence $\overline{A}_t$ close to trial $t$, and are compatible with the experimental treatment rule ("policy"). Importantly, this enables estimation of interpretable causal parameters, such as the effect of treatment on the most recent trial, $\mathbb{E}\left[Y_t(a_t = 1) - Y_t(a_t = 0)\right]$. These causal contrasts have grown popular recently in the analysis of mobile health studies [5], where they are referred to as "excursion effects." However, current methods are restricted to estimating excursion effects for the $\Delta = 1$ case in experimental designs like ours, and thus preclude estimation of effects defined only for $\Delta > 1$ (e.g., the micro longitudinal effects in Figures 1 and 4). Mobile health studies often include treatment rules with positivity violations: due to ethical or practical constraints, treatment must be withheld in certain cases (e.g., no phone notifications while driving). [5] use the notation that treatment is withheld when the time-varying "availability" indicator, $I_t$, equals zero. Similarly, in "closed-loop" optogenetics experiments, $I_t = 1$ when the conditions are met such that neural manipulation may occur (e.g., when the animal approaches the lever in the example in Section 1). There have been proposals for methods intended to account for such implied positivity violations [19, 5, 26], such as the availability-conditional estimand [5]: $\mathbb{E}\left[Y_t(a_t = 1) - Y_t(a_t = 0) \mid I_t = 1\right]$. However, estimands proposed for these settings are defined only for $\Delta = 1$. Thus, in the presence of these positivity violations, there is currently no methodology to conduct causal inference for longer proximal treatment sequences. We note that machine learning based causal methods including causal transformers [18], counterfactual recurrent networks [3], and recurrent marginal structural networks [15] are comparable to HR-MSMs that condition on all measured variables prior to the first intervention timepoint ($t - \Delta + 1$). These methods target effects of static treatment sequences and require a positivity assumption, and thus cannot be applied in closed-loop designs. They also do not provide tools for statistical inference.

## 3   Methods

To fill the gaps identified above, we propose HR-MSMs for proximal sequences of dynamic treatment regimes, designed to be compatible with treatment availability restrictions in this scientific context. These estimands are defined for any $\Delta \geq 1$, can incorporate time-varying effect modifiers, and can dissect more intricate patterns of treatment over time, compared to standard excursion effects.

### 3.1   HR-MSMs for Dynamic Treatment Regimes

Adopting the notation from [5], we define $I_t := \mathbb{1}(\mathbb{P}[A_t = 1 \mid H_t] > 0)$ as an "availability indicator", i.e., $I_t = 0$ if and only if active treatment (e.g., laser stimulation) is prohibited by design. Define $\mathcal{D}_t = \{d_t : \mathcal{H}_t \to \{0, 1\} \mid d_t(H_t) = 0 \text{ if } I_t = 0\}$, for any $t$, to be the class of treatment rules at time $t$ compatible with $I_t$. In particular, we will consider the deterministic rules $\mathcal{D}_t^* = \{d_t^{(0)}, d_t^{(1)}\} \subset \mathcal{D}_t$, where $d_t^{(0)} \equiv 0$, $d_t^{(1)} \equiv I_t$. In words, $d_t^{(0)}$ fixes $A_t = 0$, and $d_t^{(1)}$ sets $A_t$ equal to $I_t$. The treatment rules $d_t^{(0)}, d_t^{(1)} \in \mathcal{D}_t$ represent the two most extreme policies whose effects remain identifiable. We can combine these time-specific rules to construct multiple time-point analogs of excursion effects compatible with availability restrictions: for $\Delta \in \mathbb{N}$, we let $\overline{\mathcal{D}}_{\Delta,t}$ be a subset of $\mathcal{D}_{t-\Delta+1}^* \times \cdots \times \mathcal{D}_t^*$, taking $\boldsymbol{d}_{\Delta,t} = (d_{t-\Delta+1}, \ldots, d_t) \in \overline{\mathcal{D}}_{\Delta,t}$ to be a sequence of $\Delta$ treatment rules (compatible with availability restrictions) for trials $j \in \{t - \Delta + 1, ..., t\}$. The counterfactual outcome under this policy sequence is defined to be

$$Y_t(\boldsymbol{d}_{\Delta,t}) = Y_t(A_1, \ldots, A_{t-\Delta}, d_{t-\Delta+1}(H_{t-\Delta+1}), \ldots, d_t(H_t(\boldsymbol{d}_{\Delta-1,t-1}))). \tag{1}$$

That is, $Y_t(\boldsymbol{d}_{\Delta,t})$ is the counterfactual outcome under an intervention that leaves the natural value of treatment for the first $t - \Delta$ trials, then sequentially determines treatment by applying $d_{t-\Delta+j}$ to $H_{t-\Delta+j}(\boldsymbol{d}_{j-1,t-\Delta+j-1})$, for $j \in [\Delta]$, where $\boldsymbol{d}_{j-1,t-\Delta+j-1} = (d_{t-\Delta+1}, \ldots, d_{t-\Delta+j-1})$.

Letting $V_t \subseteq H_t$ be a set of effect modifiers at trial $t$, we seek to estimate $\mathbb{E}[Y_t(\boldsymbol{d}_{\Delta,t}) \mid V_{t-\Delta+1}]$, the counterfactual mean outcome, conditional on effect modifiers that are observed *before* the treatment decision of trial $t - \Delta + 1$. By construction, these estimands are identifiable under standard causal assumptions (see Section 3.2). We discuss their interpretation, and compare with existing proposals in Appendix C.1. When $\Delta > 1$ and studies have many trials, there may be many potential treatment rule sequence combinations. We thus propose to estimate effects of these interventions with an MSM on the (conditional) means of the counterfactuals (1): $m(t, \boldsymbol{d}_{\Delta,t}, V_{t-\Delta+1}; \boldsymbol{\beta}) \approx \mathbb{E}[Y_t(\boldsymbol{d}_{\Delta,t}) \mid V_{t-\Delta+1}]$, where $m$ is a fixed known function. We aim to conduct inference on the MSM parameters, $\boldsymbol{\beta}$, but we do not assume that the model is necessarily well-specified, and thus treat the MSM parameters as projections onto the working model $m$ [20, 32]:

$$\boldsymbol{\beta}_0 = \arg\min_{\boldsymbol{\beta} \in \mathbb{R}^q} \sum_{t=\Delta}^{T} \sum_{\boldsymbol{d}_{\Delta,t} \in \overline{\mathcal{D}}_{\Delta,t}} \mathbb{E}\left(h(t, \boldsymbol{d}_{\Delta,t}, V_{t-\Delta+1}) \{Y_t(\boldsymbol{d}_{\Delta,t}) - m(t, \boldsymbol{d}_{\Delta,t}, V_{t-\Delta+1}; \boldsymbol{\beta})\}^2\right), \quad (2)$$

for some fixed non-negative weight function $h$. This projection approaches lies between a fully parametric strategy, that assumes $m$ is correctly specified, and a fully nonparametric approach, that places no structure across the target causal quantities. The target $\boldsymbol{\beta}_0$ is defined as the parameter of the best fitting working model $m$ (i.e., closest in $L_2(\mathbb{P})$). In practice, the choice between considering $m$ as a working model or as a correctly specified model amounts to a trade-off between bias and variance—see the discussions in [13, 12] where analogous projection parameters are proposed.

## 3.2 Identification and Estimation

In this section, we first describe the causal assumptions under which the effects of interest are identified. We then develop an inverse probability-weighted estimator of the MSM parameters, and derive their asymptotic properties. While we focus on dynamic regime HR-MSMs below, our results also apply to static regime HR-MSMs in the case that there are no availability issues (i.e., $I_t \equiv 1$). There, the treatment rule $\boldsymbol{d}_{\Delta,t}$ reduces to a corresponding static sequence $\boldsymbol{a}_{\Delta,t}$.

For each $t$, define the treatment probability function $\pi_t(a; H_t) := \mathbb{P}[A_t = a \mid H_t]$. We make the following standard assumptions, which are expected to hold in many optogenetics designs:

**Assumption 3.1.** *Consistency:* $Y_t(\boldsymbol{d}_{\Delta,t}) = Y_t$, whenever $A_j = d_j(H_j)$, for all $j \in \{t-\Delta+1, \ldots, t\}$

**Assumption 3.2.** *Positivity: For all $t \in \{\Delta, \ldots, T\}$, and $d_t \in \mathcal{D}_t^*$, $\pi_t(d_t(H_t); H_t) \geq \epsilon$, w.p. 1*

**Assumption 3.3.** *Sequential randomization:* $A_s \perp\!\!\!\perp Y_t(\boldsymbol{d}_{\Delta,t}) \mid H_s$, for all $t \in \{\Delta, \ldots, T\}$, $s \in \{t - \Delta + 1, \ldots, t\}$

We provide a detailed discussion of these assumptions in practice in Appendix C.2. The following result says that these three assumptions are sufficient for identification of the counterfactual means $\mathbb{E}[Y_t(\boldsymbol{d}_{\Delta,t}) \mid V_{t-\Delta+1}]$, and of the MSM parameters $\boldsymbol{\beta}_0$.

**Proposition 3.4.** *Under Assumptions 3.1–3.3, we have*

$$\mathbb{E}(Y_t(\boldsymbol{d}_{\Delta,t}) \mid V_{t-\Delta+1}) = \mathbb{E}_{\mathbb{P}}\left(\prod_{j=t-\Delta+1}^{t} \frac{\mathbb{1}(A_j = d_j(H_j))}{\pi_j(A_j; H_j)} Y_t \,\middle|\, V_{t-\Delta+1}\right).$$

*Recall that $Z_i = \{\mathcal{O}_{t,i}\}_{t=1}^{T}$ is the totality of data observed on subject $i$; suppressing subject-specific index for clarity, define $\phi(Z, \cdot) : \mathbb{R}^q \to \mathbb{R}^q$ via*

$$\phi(Z, \boldsymbol{\beta}) = \sum_{t=\Delta}^{T} \sum_{\boldsymbol{d}_{\Delta,t} \in \overline{\mathcal{D}}_{\Delta,t}} h(t, \boldsymbol{d}_{\Delta,t}, V_{t-\Delta+1}) M(t, \boldsymbol{d}_{\Delta,t}, V_{t-\Delta+1}; \boldsymbol{\beta})$$

$$\times \left[\prod_{j=t-\Delta+1}^{t} \frac{\mathbb{1}(A_j = d_j(H_j))}{\pi_j(A_j; H_j)}\right] \{Y_t - m(t, \boldsymbol{d}_{\Delta,t}, V_{t-\Delta+1}; \boldsymbol{\beta})\},$$

*where $M(t, \boldsymbol{d}_{\Delta,t}, V_{t-\Delta+1}; \boldsymbol{\beta}) = \nabla_{\boldsymbol{\beta}} m(t, \boldsymbol{d}_{\Delta,t}, V_{t-\Delta+1}; \boldsymbol{\beta})$. Then, assuming the solution to (2) is unique, and the working model $m$ is differentiable in $\boldsymbol{\beta}$, the MSM parameters $\boldsymbol{\beta}_0$ are identified through the estimating equation $\mathbf{0} = \mathbb{E}_{\mathbb{P}}(\phi(Z, \boldsymbol{\beta}_0))$.*

The result of Proposition 3.4 is a population inverse probability-weighted estimating equation for the target parameters $\boldsymbol{\beta}_0$. This estimating equation motivates a corresponding IPW estimator, $\widehat{\boldsymbol{\beta}}$, solving the empirical IPW estimating equation $\mathbb{P}_n[\phi(Z, \widehat{\boldsymbol{\beta}})] = \mathbf{0}$. In the optogenetics applications of interest, the propensity scores $\pi_t$ are known by design, and can be plugged in when estimating $\widehat{\boldsymbol{\beta}}$.

We now prove asymptotic normality of the our estimator, $\widehat{\boldsymbol{\beta}}$, under mild conditions. We require the following notation: define $\boldsymbol{A}(\boldsymbol{\beta}) = \mathbb{E}[\phi(Z, \boldsymbol{\beta})\phi(Z, \boldsymbol{\beta})^T]$ and $\boldsymbol{B}(\boldsymbol{\beta}) = \mathbb{E}[\nabla_{\boldsymbol{\beta}} \phi(Z, \boldsymbol{\beta})]$.

**Theorem 3.5.** *Suppose Assumptions 3.1–3.3 and the following conditions hold:*

*(i) The minimizer $\boldsymbol{\beta}_0$ in (2) is unique;*

*(ii) $m(t, \boldsymbol{d}_{\Delta,t}, V_{t-\Delta+1}; \boldsymbol{\beta})$ is Donsker in $\boldsymbol{\beta}$, continuously differentiable at $\boldsymbol{\beta}_0$, uniformly in $V_{t-\Delta+1}$;*

*(iii) In a neighborhood around $\boldsymbol{\beta}_0$, $\boldsymbol{A}(\boldsymbol{\beta})$ and $\boldsymbol{B}(\boldsymbol{\beta})$ are finite-valued, and $\boldsymbol{B}(\boldsymbol{\beta})$ is non-singular;*

*(iv) $\widehat{\boldsymbol{\beta}} \xrightarrow{p} \boldsymbol{\beta}_0$.*

*Then $\sqrt{n}(\widehat{\boldsymbol{\beta}} - \boldsymbol{\beta}_0) \xrightarrow{d} \mathcal{N}(\mathbf{0}, \boldsymbol{V}(\boldsymbol{\beta}_0))$, where $\boldsymbol{V}(\boldsymbol{\beta}) = \boldsymbol{B}(\boldsymbol{\beta})^{-1}\boldsymbol{A}(\boldsymbol{\beta})\boldsymbol{B}(\boldsymbol{\beta})^{-1}$.*

Theorem 3.5 gives the asymptotic distribution of the estimator $\widehat{\boldsymbol{\beta}}$. The conditions (i)–(iv) are relatively mild; see Appendix C.3 for a discussion. Theorem 3.5 provides a strategy to construct asymptotically valid Wald-based confidence intervals for the MSM parameters $\boldsymbol{\beta}_0$: for any $\boldsymbol{\beta}$ we can take

$$\widehat{\boldsymbol{A}}(\boldsymbol{\beta}) = \mathbb{P}_n[\phi(Z, \boldsymbol{\beta})\phi(Z, \boldsymbol{\beta})^T], \ \ \widehat{\boldsymbol{B}}(\boldsymbol{\beta}) = \mathbb{P}_n[\nabla_{\boldsymbol{\beta}} \phi(Z, \boldsymbol{\beta})],$$

and define $\widehat{\boldsymbol{V}} = \widehat{\boldsymbol{B}}(\widehat{\boldsymbol{\beta}})^{-1}\widehat{\boldsymbol{A}}(\widehat{\boldsymbol{\beta}})\widehat{\boldsymbol{B}}(\widehat{\boldsymbol{\beta}})^{-1}$, which is consistent for $\boldsymbol{V}(\boldsymbol{\beta}_0)$. Then, for $j \in [q]$, an $(1-\alpha)$ confidence interval for $\beta_{j,0}$ is given by $\widehat{\beta}_j \pm z_{1-\alpha/2}\sqrt{\frac{\widehat{V}_{jj}}{n}}$, where $z_{1-\alpha/2}$ is the $(1-\alpha/2)$-quantile of the standard normal distribution, and $\widehat{V}_{jj}$ is the $j$-th diagonal element of $\widehat{\boldsymbol{V}}$. Confidence intervals for any linear combination of the $\boldsymbol{\beta}$ parameters can be constructed in a similar fashion.

We provide an implementation that builds the necessary dataset (with each observation copied once for every regime in $\overline{D}_{\Delta,t}$), calculates the corresponding IPW weights, and estimates the HR-MSM parameters $\boldsymbol{\beta}$ by solving the estimating equation in expression 2 using the `rootSolve` R package [11]. The process takes about 10 seconds on a standard laptop, for $> 100,000$ total (pre-copy) trials.

## 4 Experiments

### 4.1 Simulation Studies

**Experimental Setup**  We sought to assess performance of the proposed estimator, and identify variance estimators that yield nominal coverage in the small $n$ settings common in optogenetics studies. To evaluate the accuracy of our framework in estimating mean counterfactuals, we designed the simulations such that the target estimands—contrasts of mean counterfactuals—corresponded to regression coefficients from the true HR-MSM. The data were simulated to mimic closed-loop optogenetics designs with positivity violations: we drew i) $X_0 \sim \text{Bernoulli}(1/2)$; ii) $A_t \mid X_t \sim \text{Bernoulli}(\frac{1}{2}X_t)$, for $t \in \{0, \dots, T\}$; iii) $X_t \mid A_{t-1} \sim \text{Bernoulli}(0.4 + 0.4A_{t-1})$, for $t \in [T]$; and iv) $Y_t \mid X_{t-1}, A_{t-1}, X_t, A_t \sim \mathcal{N}(\alpha_1 X_{t-1} + \alpha_2 A_{t-1} + \alpha_3 X_t + \alpha_4 A_t, \sigma_t^2)$, for $t \in [T]$, where $(\alpha_1, \alpha_2, \alpha_3, \alpha_4) = (0.25, 2, 1.75, 0.5)$, and $\sigma_t^2 = 1$ for all $t$. These set availability indicator $I_t \equiv X_t$, for all $t$, and result in marginal probabilities $\mathbb{P}[X_t = 1] = \frac{1}{2}$, $\mathbb{P}[A_t = 1] = \frac{1}{4}$, for all $t$. We obtain a closed form for the parameters of the saturated two time-point dynamic treatment regime HR-MSM: letting $\boldsymbol{d}_{2,t} = (d_{t-1}, d_t) \in \overline{\mathcal{D}}_{2,t}$ be arbitrary, and defining $J_{t-1} := \mathbb{1}(d_{t-1} \equiv d_{t-1}^{(1)})$, $J_t := \mathbb{1}(d_t \equiv d_t^{(1)})$, we derive in Appendix D.1 that $\mathbb{E}(Y_t(\boldsymbol{d}_{2,t})) = \beta_0 + \beta_1 J_{t-1} + \beta_2 J_t + \beta_3 J_{t-1}J_t$, where $\beta_0 = 0.5\alpha_1 + 0.4\alpha_3$, $\beta_1 = 0.5\alpha_2 + 0.2\alpha_3$, $\beta_2 = 0.4\alpha_4$, and $\beta_3 = 0.2\alpha_4$. Aggregating $\boldsymbol{\beta} = (\beta_0, \beta_1, \beta_2, \beta_3)$, the HR-MSM given by

$$m(t, \boldsymbol{d}_{2,t}; \boldsymbol{\beta}) = \beta_0 + \beta_1 J_{t-1} + \beta_2 J_t + \beta_3 J_{t-1}J_t \tag{3}$$

is correctly specified under this data generating process, and we can evaluate the performance of the proposed estimator relative to these true values.

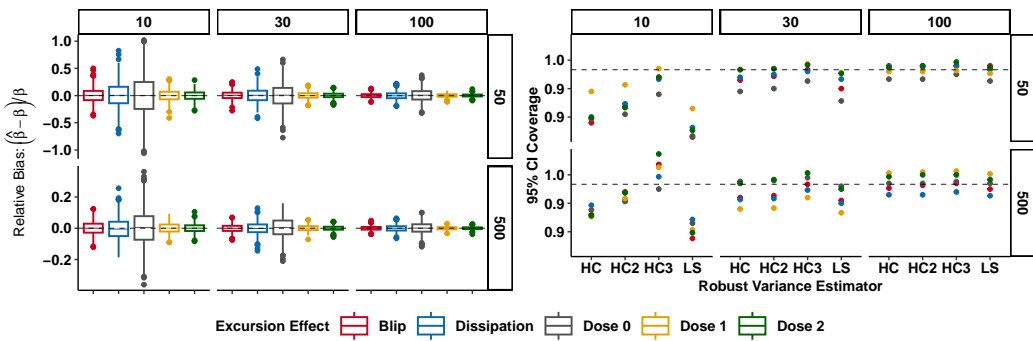

Figure 2: **Simulation Results** Panel columns indicate sample sizes, $n$ (10, 30, or 100), and panel rows indicate number of trials, $T$ (cluster sizes, 50 or 500). [Left] Relative bias associated with each sequential excursion effect. These results show that our estimator is consistent for the target parameters. [Right] 95% Confidence interval (CI) coverage for the sequential excursion effects. The coverage of 95% CIs constructed using one of three established robust variance estimators and our robust large sample (shown as *LS*) variance estimator. The nominal coverage is reached for either large $n$ or large $t$ for all estimators.

To show we can conduct valid inference on sequential excursion effects, we estimated the three estimands illustrated in Figure 1: (1) the "blip" effect of an additional exposure opportunity at the more proximal trial $t$, while keeping treatment at $t-1$ fixed at the control condition ($\beta_2 = \mathbb{E}[Y_t(d_{t-1}^{(0)}, d_t^{(1)}) - Y_t(d_{t-1}^{(0)}, d_t^{(0)})]$); (2) "effect dissipation," comparing the effect of an exposure opportunity at one versus two trials prior to the measurement of the outcome ($\beta_2 - \beta_1 = \mathbb{E}[Y_t(d_{t-1}^{(0)}, d_t^{(1)}) - Y_t(d_{t-1}^{(1)}, d_t^{(0)})]$); and (3) the "dose response" curve of exposure opportunities, where the two possible sequences for a single opportunity are averaged (the sequence $(\beta_0, \ \beta_0 + \frac{1}{2}\{\beta_1 + \beta_2\}, \ \sum_{j=0}^{3}\beta_j)$). This setup also illustrates how HR-MSMs are easily specified such that sequential excursion effects can be calculated as linear combinations of the $\boldsymbol{\beta}$ parameters.

Although the HR-MSM (3) is correctly specified, we still estimated the $\boldsymbol{\beta}$ coefficients as projection parameters. We proceeded as if we started by defining $\boldsymbol{\beta}$ as the minimizers in (2), with $V_{t-\Delta+1} = \emptyset$ (i.e., no effect modifiers), and $h(t, \boldsymbol{d}_{\Delta,t}) \equiv 1$ (i.e., constant weight function). We applied the IPW point estimator for the HR-MSM parameters, $\widehat{\boldsymbol{\beta}}$, described in Section 3.2. We assessed the coverage of 95% CIs constructed from our large sample variance estimator, and the small sample size-adjusted HC, HC2, and HC3 variance estimators (using the sandwich package in R [36]). We tested performance with sample size $n \in \{6, 10, 30, 100\}$, and number of trials $T \in \{10, 50, 500\}$.

**Results** Simulation results in Figures 2 and Appendix Figure 6 show that our estimator is unbiased for the target sequential excursion effects. We present 95% CI coverage in Table 1, and HR-MSM coefficient estimate bias and MSE in Appendix Tables 2 and 3. Sandwich estimators can yield small-sample bias [36], but in small $n$ and $T$ settings, the sample size-adjusted HC3-based CIs achieve 95% coverage. All CIs achieve 95% coverage when $n$ is large, showing that we can conduct valid inference in all settings.

## 4.2 Application: Optogenetic Study

**Data** [17] tested whether optogenetically stimulating dopamine (DA) release in the dorsolateral striatum while an animal engaged in a specific "pose" (e.g., exploring, rearing, grooming) could "teach" mice to exhibit that movement more frequently. This study was foundational in identifying the role this region plays in learning. To that end, the researchers implanted mice with optogenetics machinery, and filmed them freely-moving in a behavioral chamber. They used a pre-trained hidden Markov model to estimate an animal's pose online in real-time. They first measured the animals' target pose frequency on a baseline session without optogenetics. Then, on a subsequent treatment session, they applied the laser on a random subset of the target pose occurrences. They repeated this experiment for six target poses, in both the optogenetics and control (laser has no effect) groups.

| | | T = 50 | | | | T = 500 | | | |
|--------|-----|------------|------------|------------|------------|------------|------------|------------|------------|
| Effect | CI  | $n = 6$ | $n = 10$ | $n = 30$ | $n = 100$ | $n = 6$ | $n = 10$ | $n = 30$ | $n = 100$ |
| Blip | HC | $0.94 \pm 0.01$ | $0.94 \pm 0.01$ | $0.95 \pm 0.01$ | $0.95 \pm 0.01$ | $0.99 \pm 0.00$ | $0.97 \pm 0.01$ | $0.95 \pm 0.01$ | $0.95 \pm 0.01$ |
| | LS | $0.86 \pm 0.01$ | $0.88 \pm 0.01$ | $0.93 \pm 0.01$ | $0.95 \pm 0.01$ | $0.86 \pm 0.01$ | $0.89 \pm 0.01$ | $0.93 \pm 0.01$ | $0.94 \pm 0.01$ |
| Dissip | HC | $0.93 \pm 0.01$ | $0.94 \pm 0.01$ | $0.95 \pm 0.01$ | $0.95 \pm 0.01$ | $0.97 \pm 0.01$ | $0.96 \pm 0.01$ | $0.94 \pm 0.01$ | $0.94 \pm 0.01$ |
| | LS | $0.85 \pm 0.01$ | $0.89 \pm 0.01$ | $0.94 \pm 0.01$ | $0.95 \pm 0.01$ | $0.88 \pm 0.01$ | $0.91 \pm 0.01$ | $0.93 \pm 0.01$ | $0.94 \pm 0.01$ |
| Dose 0 | HC | $0.92 \pm 0.01$ | $0.92 \pm 0.01$ | $0.94 \pm 0.01$ | $0.94 \pm 0.01$ | $0.94 \pm 0.01$ | $0.94 \pm 0.01$ | $0.96 \pm 0.01$ | $0.95 \pm 0.01$ |
| | LS | $0.86 \pm 0.01$ | $0.88 \pm 0.01$ | $0.92 \pm 0.01$ | $0.94 \pm 0.01$ | $0.86 \pm 0.01$ | $0.91 \pm 0.01$ | $0.95 \pm 0.01$ | $0.95 \pm 0.01$ |
| Dose 1 | HC | $0.95 \pm 0.01$ | $0.95 \pm 0.01$ | $0.96 \pm 0.01$ | $0.95 \pm 0.01$ | $1.00 \pm 0.00$ | $0.97 \pm 0.01$ | $0.94 \pm 0.01$ | $0.96 \pm 0.01$ |
| | LS | $0.88 \pm 0.01$ | $0.91 \pm 0.01$ | $0.95 \pm 0.01$ | $0.95 \pm 0.01$ | $0.86 \pm 0.01$ | $0.90 \pm 0.01$ | $0.92 \pm 0.01$ | $0.96 \pm 0.01$ |
| Dose 2 | HC | $0.95 \pm 0.01$ | $0.94 \pm 0.01$ | $0.96 \pm 0.01$ | $0.96 \pm 0.01$ | $1.00 \pm 0.00$ | $0.98 \pm 0.00$ | $0.96 \pm 0.01$ | $0.96 \pm 0.01$ |
| | LS | $0.86 \pm 0.01$ | $0.89 \pm 0.01$ | $0.95 \pm 0.01$ | $0.95 \pm 0.01$ | $0.87 \pm 0.01$ | $0.90 \pm 0.01$ | $0.94 \pm 0.01$ | $0.96 \pm 0.01$ |

Table 1: **Simulation Results: CI Coverage** We achieve 95% confidence interval (CI) coverage using either small sample size-adjusted *HC3* (shown as *HC*), or our large sample (shown as *LS*) sandwich variance estimators. Mean of $R = 1000$ replicates is shown ($\pm$ standard error). We recommend *HC3* when $n$ is low. When $n$ is high, *LS* achieves nominal coverage, confirming our asymptotic theory.

To define "trials," the authors spliced the time-series of estimated pose classifications into intervals of consecutive timepoints with the same pose classification. If mice exhibited the target pose on trial $t$, they were considered "available" for optogenetic stimulation, $I_t = 1$, and were "unavailable" otherwise, $I_t = 0$. The laser was applied ($A_t = 1$) with the dynamic policy, $\mathbb{P}(A_t = 1 \mid I_t) = 0.75 I_t$. Denoting $Y_t^0$ and $Y_t$ as a binary indicator that an animal engaged in the target pose on trial $t$ of the *baseline* and *treatment* sessions, respectively, the authors estimated treatment effects of the form $\psi = \left(\mathbb{E}[\bar{Y}^1 \mid G = 1] - \mathbb{E}[\bar{Y}^0 \mid G = 1]\right) - \left(\mathbb{E}[\bar{Y}^1 \mid G = 0] - \mathbb{E}[\bar{Y}^0 \mid G = 0]\right)$ where $\bar{Y}^0 = \sum_{t=1}^{T_0} Y_t^0$, $\bar{Y}^1 = \sum_{t=1}^{T} Y_t$, and $T, T_0 \in \mathbb{N}$ are the trial numbers in treatment and baseline sessions, respectively.[4] There were $n_1 = 28$ and $n_0 = 12$ animals in the optogenetics and control groups, respectively. $T$ ranged across animals/sessions from 1207-4876, with a mean of 3612 and IQR = $[3341, 3940]$. The authors reported a (pooled across target poses) positive optogenetics treatment effect estimate akin to $\widehat{\psi}$, suggesting DA stimulation causes an increase in target pose frequency.

We argue this analysis procedure leaves many scientific questions untested. Conceptualizing optogenetics like a "study drug," we question whether stimulation immediately "taught" the animal the target pose, or whether the treatment effect on learning had a lagged onset. Similarly, did the effect of a single stimulation persist or dissipate across trials? Did more treatments lead to more learning monotonically, or is there an antagonistic effect or non-monotonic dose-response curve in learning?

**Application Methods** We applied our framework to provide a nuanced trial-by-trial characterization of the causal effects of DA stimulation, and formally answer the questions above. Specifically, we tested the causal effect of specific sequences of deterministic dynamic policies, $\boldsymbol{d}_{\Delta,t}$ (occurring on trials $t \in \{t - \Delta + 1, \ldots, t\}$), on the mean counterfactual $\mathbb{E}[Y_t(\boldsymbol{d}_{\Delta,t})]$. We defined the outcome $Y_t$ as an indicator that the mouse exhibited the target pose on trial $t + 2$, the next trial on which mice could exhibit the target pose if they were available for stimulation on trial $t$. We fit a set of HR-MSMs illustrating that we can reliably estimate the types of excursion effects in Figure 1. We describe the models we fit below, and relegate code, data and pre-processing details to Appendix Section F.

**Results** Our first question was whether standard methods reveal significant treatment effects when assessed with the estimands commonly tested in optogenetics studies. We applied a GEE with mean model, $\log\left(\mathbb{E}[\bar{Y}^s \mid G = g, S = s]\right) = \gamma_0 + \gamma_1 g + \gamma_2 s + \gamma_3 g \times s$, where $S \in \{0, 1\}$ indicates baseline and optogenetics sessions, respectively. The estimate $\widehat{\gamma}_3$, shown in Figure 3F, thus provides a treatment effect estimate for the *observed* stochastic dynamic policy in [17]. We adopted a Poisson working model, since [17] analyzed $\bar{Y}^1, \bar{Y}^0 \in \mathbb{N}$. We tested these (macro longitudinal) effects for each pose individually, rather than pooling over them, as in [17]. The model yielded no significant effects for any individual pose. We show boxplots in Appendix Figure 10 of the subject-level summary $\bar{Y}^1 - \bar{Y}^0$ that is compared across groups in this model. Outcome levels are similar across groups for most poses, further highlighting how standard outcome summaries can obscure effects.

To assess an analogous "local" treatment effect using our method, we tested the impact of a single stimulation opportunity. We further evaluated whether the effect had a lagged onset and/or dissipated

---

[4]The authors used a Mann Whitney U Test applied to a summary across poses but, in keeping with the mean counterfactual-based causal estimands, we describe it in terms of means (not medians) and individual poses.

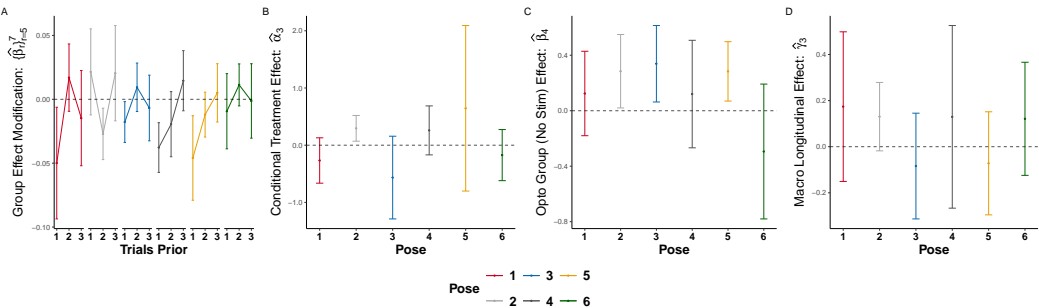

Figure 3: **Optogenetics Analyses.** Plots show coefficient estimates (error bars show 95% CIs). Columns/colors indicate the target pose. [A] Interaction term between G and sequential excursion effect of a single "dose" occuring $r = 1, 2,$ or $3$ trials prior to the proximal outcome that the "dose" occurred on. The excursion effects are significant for poses 1-5 (at, at least, one lag level). [B] Availability conditional estimate of interaction $A \times G$: laser $\times$ group interaction. [C] Main effect of G under a "no-recent-treatment opportunity" policy; this reflects the average causal effect of group among a population that has received no laser opportunities in the last $\Delta = 3$ trials. [D] Macro longitudinal analysis, similar to original paper, identifies no significant effects.

across trials. We included group, $G$, as an effect modifier, to test whether the causal effect of this treatment opportunity was larger in one of the groups. Setting $\Delta = 3$, and restricting the regimes of interest to those with at most one treatment opportunity "dose", $\boldsymbol{d}_{3,t} \in \{(d_{t-2}, d_{t-1}, d_t) : \sum_{j=t-2}^{t} \sigma_j(d_j) \leq 1\}$, where $\sigma_j(d_j) = \mathbb{1}(d_j = d_j^{(1)})$, we fit the HR-MSM

$$\text{logit}\left(\mathbb{E}[Y_t(\boldsymbol{d}_{\Delta,t}) \mid G = g]\right) = \beta_0 + \sum_{r=0}^{2} \beta_{r+1}\sigma_{t-r}(d_{t-r}) + \beta_4 g + \sum_{r=0}^{2} \beta_{5+r} g \times \sigma_{t-r}(d_{t-r}). \quad (4)$$

Thus, $\widehat{\beta}_r$ with $r \in [3]$ is an estimate of the log odds ratio comparing the mean counterfactual of $Y_t$ under a treatment sequence with a single dose (on $r = 1, 2,$ or $3$ trials prior) vs. a treatment sequence with zero dose. This permits assessment of effect dissipation or persistence. Figure 1B illustrates the analogous effect under a static regime. The interaction terms, $\widehat{\beta}_r$ with $r \in \{5, 6, 7\}$, quantify how these causal effects of a recent treatment opportunity differ between the two groups.

The results from our model (4) reveal that stimulation opportunities in the treatment group tend to *reduce* the odds of the outcome, compared to the control group. As shown in Figure 3C, these effects are significantly negative for at least one lag level in five out of six target poses. In personal communications, the authors of [17] stated that this result appeared consistent with their finding that animal exploration increased right after stimulation (quantified as higher pose entropy). Figure 3D shows the main effect of group under a treatment sequence of dose zero. In essence, this provides an estimate of the "long-term" effect of DA stimulation: $\widehat{\beta}_4$ is the log odds ratio of treatment group under a regime of dose zero (i.e., a "no recent stimulation" policy). We fit comparable models for sequences as long as $\Delta = 7$ and found results were similar across $\Delta$ values.

Next, we fit the analogous model for the availability-conditional estimand [5] to determine whether current excursion effect methods (i.e., those confined to $\Delta = 1$ policies) identify the same treatment effects: $\text{logit}\left(\mathbb{E}[Y_t(a_t) \mid I_t = 1, G = g]\right) = \alpha_0 + \alpha_1 a_t + \alpha_2 g + \alpha_3 g \times a_t$. Figure 3C shows that the effect estimates, $\widehat{\alpha}_3$, are significant in only one pose. These results highlight how our approach can uncover a greater number of significant effects.

Finally, in Appendix Section E, we include analyses showing that the laser exhibits a dose-response curve in both groups: more treatment opportunities (on some poses) in the last $\Delta = 5$ trials causes the animal to exhibit the target pose more often. Additionally, there is significant effect modification by baseline responding: the laser has a larger effect in animals who exhibited high baseline pose frequency. Together, these results show we can reliably estimate sequential excursion effects.

## 5 Discussion

We propose the first, to the best of our knowledge, formal causal inference framework for closed-loop optogenetics behavioral studies. We introduce a nonparametric excursion effect framework, an associated IPW estimator (with valid CIs), with a scalable implementation, and proved its consistency

and asymptotic normality under mild assumptions. Methodologically, our proposed sequential excursion effects represent an expansion of the conditional estimands proposed in [5] to longitudinal policies ($\Delta \geq 1$), in the presence of positivity violations. Our methods also directly apply to "open-loop" (static policies) designs, as they arise as a special case when $I_t = 1$ for all $t \in [T]$.

HR-MSMs are powerful and useful models, but have their limitations. As has been discussed in the causal inference literature, these estimands marginalize over all treatments for trials $t \in [t - \Delta]$, and thus depend on the protocol used in the design [31, 6]. Moreover, while contrasts of our estimands are null under the sharp null of no causal effect of treatment (e.g., optogenetic laser stimulation), effects should generally still be interpreted in terms of treatment *opportunities*. Finally, while our implementation is computationally efficient, we anticipate computational challenges for very large $\Delta$.

The model $m$ and the number of intervention timepoints $\Delta$ represent key choices for practitioners. Our inferential results (i.e., Theorem 3.5) are valid for a large class of working mean models $m$, and notably do not rely on any distributional assumptions. The "Donsker" requirement (condition (ii) in Theorem 3.5) is satisfied outright by generalized linear models such as those we use in our application [35], as well as some formulations of random forests [34] and kernel estimators [2]. In future work, we will study how inference can be obtained for more flexible models that do not satisfy the Donsker assumption. Likewise, the value of $\Delta$ plays a significant role as it determines the nature of the effects being estimated, and should be chosen on the basis of subject matter expertise. That said, we found that 2-3 intervention timepoints are often sufficient to capture a rich set of sequential excursion effects, and in our application that results were relatively stable across a range of $\Delta$ values.

The application highlights the drawbacks of standard optogenetics analysis methods. Our finding that macro longitudinal estimates of individual target poses show almost no effect between groups highlights how "treatment–confounder" feedback can obscure strong treatment effects in closed-loop designs, even when inspecting simple averages of observed outcomes. Our methods account for this by careful causal adjustment with IPW. In personal communications with the authors of [17], they agreed with our findings and remarked at how these methods reveal a collection of causal effects that are difficult to uncover without sophisticated causal inference methods.

Our analyses reveal immediate *negative* effects (detectable on the next trial) and *positive* slower effects of DA stimulation (i.e., in treatment relative to control animals). We also find the control group exhibits positive, off-target effects of the laser. Together the *opposing* signs of these 'fast'/"slow" and on/off-target causal effects may further dilute the magnitude of macro longitudinal effects that summarize the outcome across many trials (e.g., total pose counts). Finally, by enabling estimation of *sequential* excursion effects (i.e., $\Delta > 1$), we can reveal effect profiles (e.g., dose-response curves) not possible with availability-conditional estimands whose definition is confined to $\Delta = 1$ regimes. As we observed, the optogenetics group sometimes exhibits an excursion effect not present in the control group. Thus, by combining different sequential excursion effects, analysts can, for example, disentangle laser on-target from off-target effects. When off-target effects are not a major concern, our framework enables estimation of causal effects *without* having to collect data in a control group, thereby potentially reducing the number of animals required in a study.

Although we focus on optogenetics here, our proposed methods are relevant for a wide range of mobile health, neuroscience and psychology experiments for which the "local/micro" longitudinal structure is of scientific interest. Indeed, "closed-loop" designs are common in many behavioral studies in human neuroimaging and cognitive sciences (e.g., when stimuli are conditionally randomized). We hope our methods constitute a useful methodological contribution to the causal inference literature, and will help applied researchers exploit the rich information contained in their experiments.

## Acknowledgements

This research was supported by the Intramural Research Program of the National Institute of Mental Health (NIMH), project ZIC-MH002968. This study utilized the high-performance computational capabilities of the Biowulf Linux cluster at the National Institutes of Health, Bethesda, MD (http://biowulf.nih.gov). First, we would like to thank the authors of "Spontaneous behaviour is structured by reinforcement without explicit reward," Drs. Jeffrey Markowitz and Sandeep Robert Datta, for sharing their data, helping us conduct analyses, and interpret the results. This work would not have been possible without their generosity, commitment to open science and scientific rigor. We would also like to thank the NIMH Machine Learning Team for helpful feedback on our project.

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

## A   Micro Longitudinal Effects

In Appendix Figure 4, we illustrate additional micro longitudinal effects that can be probed with our sequential excursion effect framework. This figure has the same layout as Figure 1A-C.

## B   Illustration of Treatment-Confounder Feedback

We provide in this section a synthetic example in which two independent groups exhibit identical mean outcome patterns over time, but where the treatment (e.g., turning on laser in the brain) has a substantial effect in one group but not the other. As our construction will demonstrate, this phenomenon manifests due to treatment-confounder feedback leading to effects canceling out. In a similar fashion, one can similarly construct scenarios where effects are exaggerated.

Suppose $G \in \{0, 1\}$ represents an experimentally manipulable marker (e.g., animals expressing opsin in the brain), and counterfactual outcomes under $G = g$ are denoted $Y_t^g$. We will suppose that potential outcomes generated in the active setting ($G = 1$) are given by

$$Y_t^1 \sim \mathcal{N}(\gamma_{0t} + \gamma_1 X_{t-1} + \gamma_2 A_{t-1} + \gamma_3 X_t + \gamma_4 A_t, \sigma_t^2),$$

and potential outcomes in the control condition ($G = 0$) are given by

$$Y_t^0 \sim \mathcal{N}(\gamma_{0t} + \gamma_1 X_{t-1} + \gamma_3 X_t, \sigma_t^2),$$

i.e., the treatment (e.g., laser) has an effect when $G = 1$, but not when $G = 0$.

Suppose further that a behavior $X_t$ is measured at all time points $t$, and determines whether or not treatment will be administered with positive probability. Like the outcomes, this behavior will be affected by the laser only when $G = 1$:

$$X_t^g \sim \text{Bernoulli}(0.7 - 0.5 \, A_{t-1} \, g), \text{ for } t \in \{1, \dots, T\},$$

and $X_0^g \sim \text{Bernoulli}(\frac{1}{2})$ at baseline.

Now we consider a study where animals are randomly assigned at baseline to either $G = 1$ or $G = 0$. At each time point $t$, the behavior $X_t$ is measured, and treatment is then drawn according to $A_t \sim \text{Bernoulli}(0.8 X_t)$. By induction, $\mathbb{E}(A_t \mid G = 1) = 0.4$ and $\mathbb{E}(X_t \mid G = g) = 0.7 - 0.2g$, for all $t$. It follows that

$$
\begin{aligned}
&\mathbb{E}(Y_t \mid G = g) \\
&= \gamma_{0t} + \gamma_1 \mathbb{E}(X_{t-1} \mid G = g) + \gamma_2 \mathbb{E}(A_{t-1} \mid G = 1)g + \gamma_3 \mathbb{E}(X_t \mid G = g) + \gamma_4 \mathbb{E}(A_t \mid G = 1)g \\
&= \{\gamma_{0t} + 0.7(\gamma_1 + \gamma_3)\} + \{-0.2(\gamma_1 + \gamma_3) + 0.4(\gamma_2 + \gamma_4)\}g.
\end{aligned}
$$

Thus, the "macro"/"global" between-group mean difference trajectory is given by

$$\mathbb{E}(Y_t \mid G = 1) - \mathbb{E}(Y_t \mid G = 0) = -0.2(\gamma_1 + \gamma_3) + 0.4(\gamma_2 + \gamma_4),$$

which will be null if $\gamma_2 + \gamma_4 = 0.5(\gamma_1 + \gamma_3)$. Notice that this cancellation is possible even if the immediate effect of treatment on the outcome is quite strong, say if $\gamma_2$ and $\gamma_4$ are large and positive. The cancellation is made possible through the opposing effects of treatment on the intermediate behavior and the outcome: when $G = 1$, $A_{t-1}$ negatively impacts $X_t$ but positively impacts $Y_t$. More generally, these $X_t$-$A_t$ feedback loops can lead to dilution or exaggeration of the actual effect of treatments when only analyzing observed mean outcomes.

We note that in the data generating scenario described in this appendix, the proposed dynamic treatment regime HR-MSM methodology would pick out non-null effects of treatment within the active group ($G = 1$), and show differing effects between groups, even if the condition above held such that observed mean outcomes were identical. This example thus serves to illustrate both the challenges with closed-loop designs and, despite these challenges, the ability of the proposed methodology to elucidate effects.

**Example Analysis on Synthetic Data**   To illustrate the above, we provide an example on a simulated dataset, taking $n = 100$, $T = 500$, $\gamma_1 = \gamma_3 = 1$, $\gamma_2 = \gamma_4 = 0.5$.

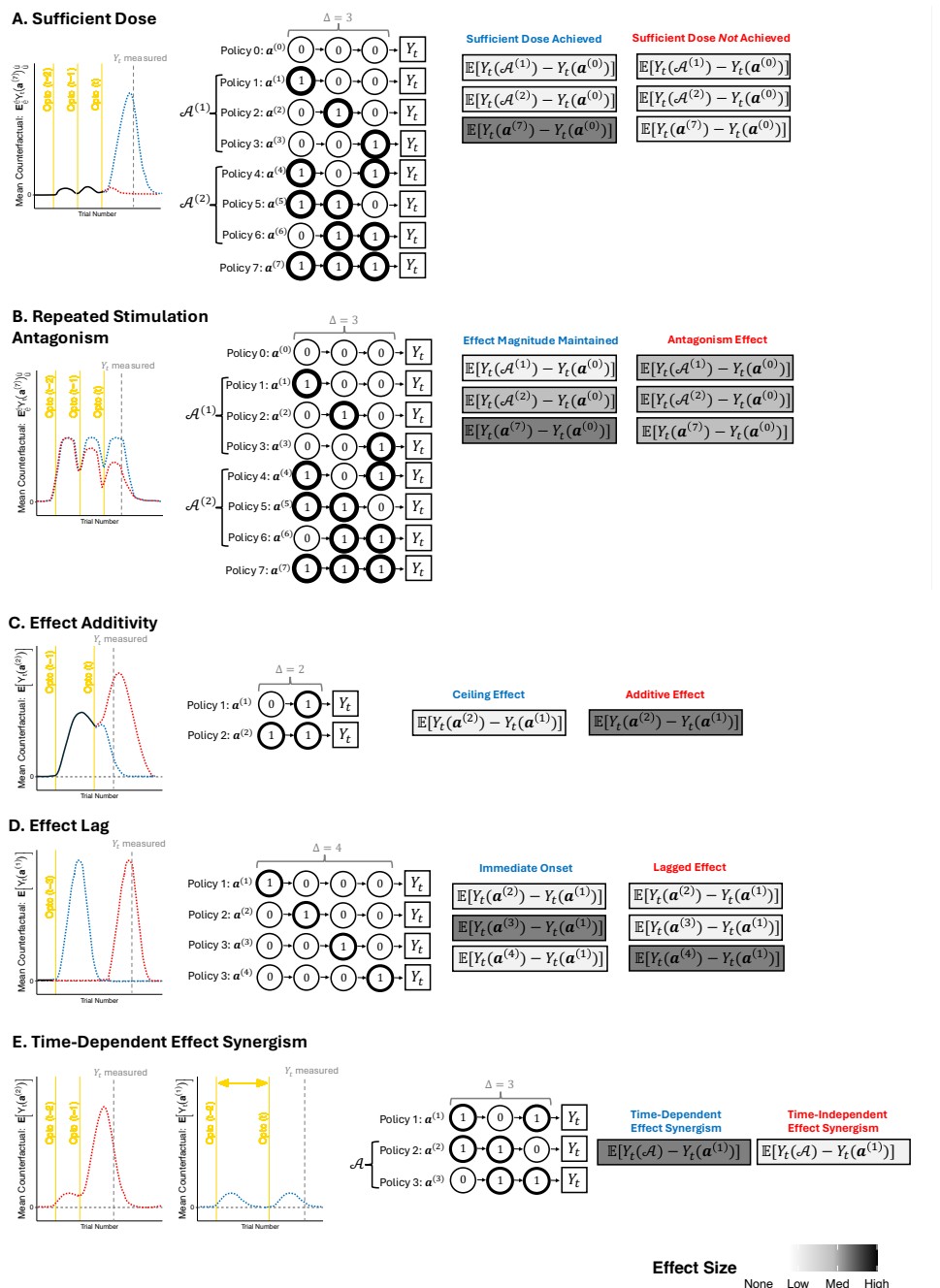

Figure 4: **Example *Sequential Excursion Effects*.** The left panels show one setting where a sequence of laser simulations do or do not have the indicated effect on the outcome. The middle panel shows deterministic static policies that could be used to construct a causal contrast to probe the effect. The right panel shows what the anticipated effect size (darker is larger) of those contrasts might be if there is or is not the indicated effect profile. [A] Sufficient dose. The red line shows how three successive stimulations is required to trigger a large effect, whereas the effect profile in blue shows that the sufficient dose has not been reached. [B] Repeated stimulation anatagonism. The red line shows a negative dose-response, and the blue line shows a stable effect size. [C] Effect additivity. The red line shows a second stimulation triggers a larger response, whereas the blue shows that the second stimulation does not increase the response substantially beyond that of the first stimulation. [D] Effect Lag. The red line shows that the causal effect of stimulation is not visible until after a lag period. The blue line shows a setting where the effect is immediate. [E] Time-dependent effect synergism. The red line shows a setting where the effect is additive provided the stimulations occur close enough together (red line), but if stimulations occur far apart, this synergism does not occur (blue line).

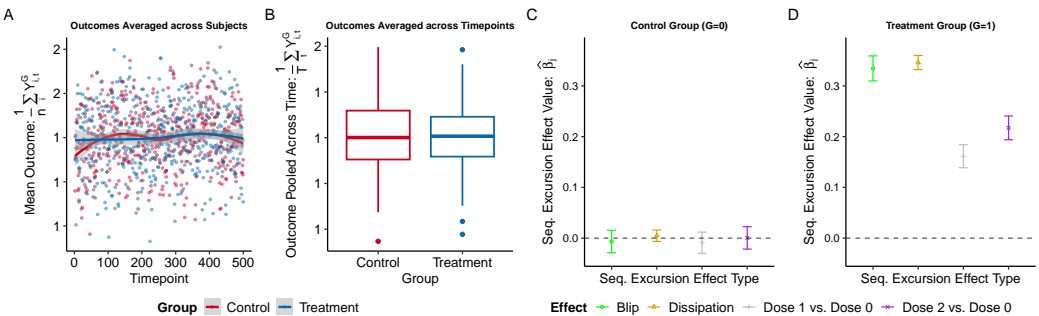

Figure 5: **Treatment–Confounder Feedback Example** Sequential excursion effects reveal causal effects obscured in "macro" summaries. Analysis results from a simulated dataset following the argument above (in Appendix B), taking $n = 100$, $T = 500$, $\gamma_1 = \gamma_3 = 1$, $\gamma_2 = \gamma_4 = 0.5$. [A] Each dot is an outcome value, $Y_{i,t}^G$, for subject $i$ at timepoint $t$ from "control" ($G = 0$), or from "treatment" ($G = 1$) groups. Lines are timepoint-specific means (averaged across subjects), estimated using a linear smoother (`loess`). (B) Same data as (A), but each point in boxplot is a subject's mean outcome value (averaged across timepoints). In (A)-(B), "macro" summaries show no differences due to treatment–confounder feedback: mean outcome values (averaged across subjects or timepoints) are nearly identical in both groups. (C)-(D) Point estimates and 95% CIs (error bars) of sequential excursion effects reveal "local" causal effects (in Treatment group only), obscured in "macro" summaries (shown in (A)-(B)).

# C  Additional Details for Section 3

## C.1  Interpretation of Sequential Excursion Effects

The interpretation of the mean counterfactual quantity $\mathbb{E}[Y_t(\boldsymbol{d}_{\Delta,t}) \mid V_{t-\Delta+1}]$ is somewhat subtle, and warrants further discussion. When $\Delta = 1$, we can express a contrast of these estimands in terms of the effect of exposure in a certain subgroup:

$$\mathbb{E}[Y_t(d_t^{(1)}) \mid V_t] - \mathbb{E}[Y_t(d_t^{(0)}) \mid V_t] = \mathbb{E}[Y_t(a_t = 1) - Y_t(a_t = 0) \mid V_t, I_t = 1]\mathbb{P}[I_t = 1 \mid V_t].$$

That is, the mean contrast in counterfactual outcomes for $d_t^{(1)}$ versus $d_t^{(0)}$ is the mean effect of $A_t = 1$ versus $A_t = 0$ among those with $I_t = 1$—the availability-conditional estimand proposed by [5]—diluted by the probability of availability. Even in this case with $\Delta = 1$, it may not always be clear for whom the availability-conditional estimand generalizes to, i.e., the group $I_t = 1$ may be highly idiosyncratic and not of particular interest. On the other hand, the parameters we are proposing summarize the effects of plausible interventions on the whole population, acknowledging that for some individuals active treatment (i.e., $A_t = 1$) is not possible.

The comparison just described is somewhat akin to the duality in clinical trials of per-protocol (or complier-specific) effects, and intention-to-treat effects. Thus, in practice when $\Delta = 1$, we would recommend assessing both the availability-conditional estimand, as in [5], as well as our proposed population-level effect. When $\Delta > 1$, it is not clear whether an analogous availability-conditional estimand exists; our approach is viable for arbitrary $\Delta$. In general, our estimands have the population-level (possibly conditional on effect modifiers) interpretation of summarizing how outcomes would be affected if the experimental protocol were changed to match $\boldsymbol{d}_{\Delta,t}$ for the $\Delta$ trials leading up to the outcome. Finally, we note that, as for all excursion effects or history-restricted marginal structural models, the estimands under study are dependent on the treatment protocol [6].

## C.2  Discussion of Causal Assumptions

Consistency (Assumption 3.1) states that for any of the regimes $\boldsymbol{d}_{\Delta,t}$ under study, the counterfactual outcome $Y_t(\boldsymbol{d}_{\Delta,t})$ equals the observed outcome $Y_t$ when observed treatment values correspond to assignment under $\boldsymbol{d}_{\Delta,t}$. Positivity (Assumption 3.2) states that treatment probabilities are bounded away from zero—this is required for the asymptotic analysis of the proposed estimator later on. Note that, by definition of the availability indicator $I_t$, and the regimes $\mathcal{D}_t^*$ in Section 3.1, we are allowing $\mathbb{P}[A_t = 1 \mid H_t] = 0$ in some cases (i.e., when $I_t = 0$), but Assumption 3.2 rules out $\mathbb{P}[A_t = 1 \mid H_t] = 1$. This positivity assumption holds in many open- and closed-loop optogentic studies. In practice, in such experiments, one can ensure that Assumption 3.2 holds by design when choosing the treatment assignment probabilities. Finally, Assumption 3.3 says that treatments are randomly assigned at each time $t$, based on all previously measured data $H_t$. In the sequential optogenetic experiments that motiviate this work, this assumption would hold by design. In observational studies, one will have to assess the plausibility of Assumption 3.3 (as well as Assumptions 3.1 and 3.2) on a case-by-case basis, ideally based on subject matter knowledge; it may be harder to justify Assumption 3.3 due to the possible presence of unmeasured confounders.

## C.3  Conditions of Theorem 3.5

Conditions (i) through (iv) are standard conditions for asymptotic normality of M-estimators [8, 9]. For condition (ii), we expect the working model $m$ to be differentiable in $\boldsymbol{\beta}$ for most common models. Moreover, for standard generalized linear models, $m$ will be appropriately Donsker—see [35] for formal definitions. Condition (iii) is satisfied under mild conditions, e.g., if the weight functions $h$, the model $m$ and its derivative $M$, and the outcomes $Y_t$ are uniformly bounded, and no haphazard degeneracy in $\boldsymbol{B}$ exists that could cause singularity. Lastly, condition (iv) is also quite weak, only requiring convergence of $\widehat{\boldsymbol{\beta}}$ at an arbitrarily slow rate, and would hold under some stochastic equicontinuity conditions [22, 24].

## C.4 Proofs of results in Section 3.2

*Proof of Proposition 3.4.* This result follows the usual $g$-formula identification argument [27]: defining $\boldsymbol{A}_{\Delta,t} = (A_{t-\Delta+1}, \ldots, A_t)$, $\boldsymbol{d}_{\Delta,t}(H_t)) = (d_{t+\Delta+1}(H_{t+\Delta+1}), \ldots, d_t(H_t))$,

$$\mathbb{E}(Y_t(\boldsymbol{d}_{\Delta,t}) \mid V_{t-\Delta+1})$$
$$= \mathbb{E}(\mathbb{E}(Y_t(\boldsymbol{d}_{\Delta,t})) \mid H_{t-\Delta+1}) \mid V_{t-\Delta+1})$$
$$= \mathbb{E}(\mathbb{E}(Y_t(\boldsymbol{d}_{\Delta,t})) \mid H_{t-\Delta+1}, A_{t-\Delta+1} = d_{t-\Delta+1}(H_{t-\Delta+1})) \mid V_{t-\Delta+1})$$
$$\cdots$$
$$= \mathbb{E}(\mathbb{E}(\cdots \mathbb{E}(Y_t(\boldsymbol{d}_{\Delta,t}) \mid H_t, \boldsymbol{A}_{\Delta,t} = \boldsymbol{d}_{\Delta,t}(H_t)) \cdots \mid H_{t-\Delta+1}, A_{t-\Delta+1} = d_{t-\Delta+1}(H_{t-\Delta+1})) \mid V_{t-\Delta+1}),$$

where we repeatedly invoke iterated expectations and Assumption 3.3 (justified by Assumption 3.2), then use Assumption 3.1 in the last equality. We can then rewrite this formula in an equivalent IPW form:

$$\mathbb{E}(\mathbb{E}(\cdots \mathbb{E}(Y_t(\boldsymbol{d}_{\Delta,t}) \mid H_t, \boldsymbol{A}_{\Delta,t} = \boldsymbol{d}_{\Delta,t}(H_t)) \cdots \mid H_{t-\Delta+1}, A_{t-\Delta+1} = d_{t-\Delta+1}(H_{t-\Delta+1})) \mid V_{t-\Delta+1})$$
$$= \mathbb{E}\left(\mathbb{E}\left(\frac{\mathbb{1}(A_{t-\Delta+1} = d_{t-\Delta+1}(H_{t-\Delta+1}))}{\pi_{t-\Delta+1}(A_{t-\Delta+1}; H_{t-\Delta+1})} \cdots \mathbb{E}\left(\frac{\mathbb{1}(A_t = d_t(H_t))}{\pi_t(A_t; H_t)} Y_t(\boldsymbol{d}_{\Delta,t}) \mid H_t\right) \cdots \mid H_{t-\Delta+1}\right) \mid V_{t-\Delta+1}\right)$$
$$= \mathbb{E}\left(\prod_{j=t-\Delta+1}^{t} \frac{\mathbb{1}(A_j = d_j(H_j))}{\pi_j(A_j; H_j)} Y_t \mid V_{t-\Delta+1}\right),$$

where the last equality is achieved again by iterated expectations. The second statement in Proposition 3.4 is obtained by differentiating (2) with respect to $\boldsymbol{\beta}$, setting this to zero, then invoking the first statement of Proposition 3.4 (which we have just proved). $\square$

*Proof of Theorem 3.5.* This is an immediate application of Theorem 5.31 in [35]. $\square$

# D  Additional Simulation Details and Results

## D.1  HR-MSM for Simulation Data-Generating Mechanism

In this section, we derive the form of the HR-MSM in (3), and show that it is implied by the data-generating mechanism of the simulation study. First, observe that for $t \geq 2$,

$$Y_t(d_{t-1}, d_t) = \alpha_1 X_{t-1} + \alpha_2 d_{t-1}(X_{t-1}) + \alpha_3 X_t(d_{t-1}) + \alpha_4 d_t(X_t(d_{t-1})) + \epsilon_t,$$

for some exogenous $\epsilon_t \sim \mathcal{N}(0, \sigma_t^2)$, where $X_t(d_{t-1})$ is the potential $X_t$ value under the intervention setting $A_{t-1}$ to $d_{t-1}(X_{t-1})$. Note that $d_{t-1}(X_{t-1}) = J_{t-1} X_{t-1}$, and by our structural equations, $X_t(d_{t-1}) \sim \text{Bernoulli}(0.4 + 0.4 d_{t-1}(X_{t-1}))$, so that $\mathbb{E}(X_t(d_{t-1})) = 0.4 + 0.2 J_{t-1}$, recalling that $\mathbb{E}(X_{t-1}) = 0.5$. Finally, $d_t(X_t(d_{t-1})) = J_t \cdot \text{Bernoulli}(0.4 + 0.4 d_{t-1}(X_{t-1}))$, which gives $\mathbb{E}(d_t(X_t(d_{t-1}))) = \{0.4 + 0.2 J_{t-1}\} J_t$. Putting everything together, we obtain

$$\mathbb{E}(Y_t(d_{t-1}, d_t)) = 0.5\alpha_1 + 0.5 J_{t-1}\alpha_2 + \{0.4 + 0.2 J_{t-1}\}\{\alpha_3 + J_t\alpha_4\}$$
$$= \{0.5\alpha_1 + 0.4\alpha_3\} + \{0.5\alpha_2 + 0.2\alpha_3\} J_{t-1} + 0.4\alpha_4 J_t + 0.2\alpha_4 J_{t-1}J_t$$
$$\equiv \beta_0 + \beta_1 J_{t-1} + \beta_2 J_t + \beta_3 J_{t-1}J_t,$$

as claimed.

## D.2  Further Simulation Results

In Appendix Figure 6 we present the same results as in Figure 2 but with more sample sizes, $n$ and trials, $T$. We also present these same simulation results in terms of the $\boldsymbol{\beta}$ coefficients of HR-MSM 3. In the main text we presented results in terms of sequential excursion effect parameters, which are linear combinations of these HR-MSM regression coefficients.

# E  Additional Application Results

## E.1  Dose-Response Excursion Effects

**History-Restricted MSM**  We fit an HR-MSM within the treatment group ($G = 1$) to estimate the causal effect of "dose," the number of treatment opportunities in the previous $\Delta = 5$ trials:

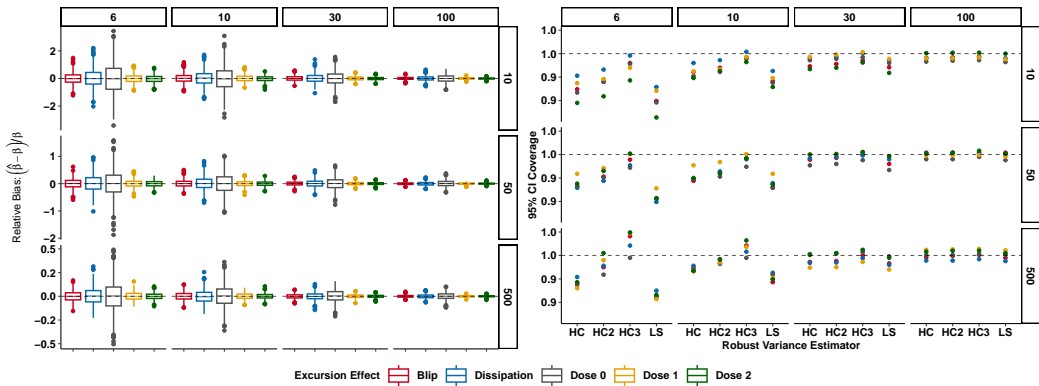

Figure 6: **Simulation Results** Panel columns indicate sample sizes, $n$, and panel rows indicate number of trials, $T$ (cluster sizes). [Left] Relative bias associated with each sequential excursion effect. These results show that our estimator is consistent for the target parameters. [Right] 95% Confidence interval (CI) coverage for the the sequential excursion effects. The coverage of 95% CIs constructed using one of three established robust variance estimators and our robust large sample (shown as *LS*) variance estimator. The nominal coverage is reached for either large $n$ or large $t$ for all estimators.

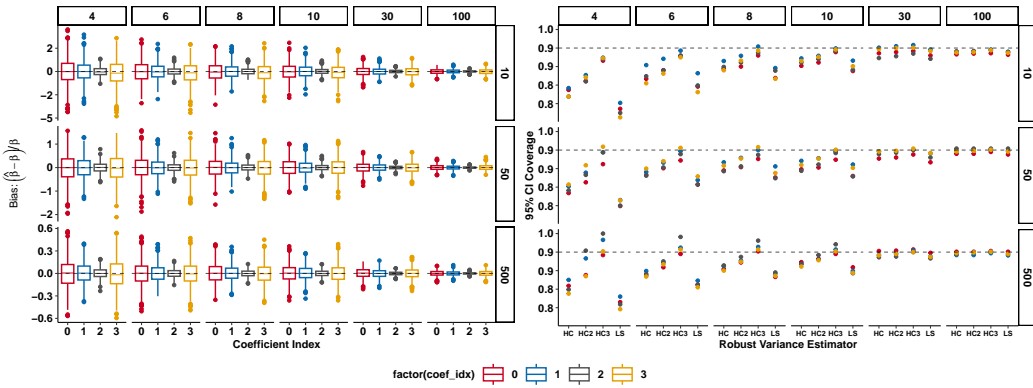

Figure 7: **Simulation Results in Terms of MSM Coefficients** Relative bias and 95% Confidence Interval (CI) coverage of regression coefficients of history-restricted marginal structural model (MSM) 3. Panel columns indicate sample sizes, $n$, and panel rows indicate number of trials, $T$ (cluster sizes). [Left] Relative bias associated with each HR-MSM regression coefficient. These results show that our estimator is consistent for the target parameters. [Right] 95% CI coverage for the MSM coefficients. The coverage of 95% CIs constructed using one of three established robust variance estimators and our robust large sample (shown as *LS*) variance estimator. The nominal coverage is reached for either large $n$ or large $t$ for all estimators.

$$\text{logit}\left(\mathbb{E}[Y_t(\boldsymbol{d}_{\Delta,t}) \mid G = 1]\right) = \beta_0 + \sum_{r=1}^{3} \beta_r \mathbb{1}\left(\sum_{j=t-\Delta+1}^{t} \sigma_j(d_j) = r\right), \tag{5}$$

where $\sigma_j(d_j) = \mathbb{1}(d_j = d_j^{(1)})$. The coefficient $\widehat{\beta}_r$ is an estimate of the log odds ratio comparing the mean counterfactual of $Y_t$ for a treatment sequence of dose $r \in [3]$ compared to a sequence of dose zero (see Figure 1C for an illustration of the static regime analogue). A dose of three is the maximum feasible dose for $\Delta = 5$ since the same pose cannot occur on two consecutive trials.

The dose-response effect estimates, $\{\widehat{\beta}_r\}_{r=1}^{3}$, from HR-MSM (5) are shown in Figure 8A. This illustrates the capacity of our approach to identify a clear dose-response effect: within the past $\Delta = 5$ trials, each additional opportunity for a stimulation *causes* an increase in the odds of engaging in the

|         | T = 50 | | | | T = 500 | | | |
|---------|--------|--------|--------|--------|--------|--------|--------|--------|
| Effect  | $n = 6$ | $n = 10$ | $n = 30$ | $n = 100$ | $n = 6$ | $n = 10$ | $n = 30$ | $n = 100$ |
| Blip    | $1.91 \pm 0.09$ | $1.10 \pm 0.05$ | $0.30 \pm 0.02$ | $0.03 \pm 0.01$ | $0.11 \pm 0.01$ | $0.03 \pm 0.01$ | $0.00 \pm 0.00$ | $0.00 \pm 0.00$ |
| Dissip  | $2.11 \pm 0.09$ | $1.23 \pm 0.06$ | $0.33 \pm 0.02$ | $0.04 \pm 0.01$ | $0.11 \pm 0.01$ | $0.03 \pm 0.01$ | $0.00 \pm 0.00$ | $0.00 \pm 0.00$ |
| Dose 0  | $0.73 \pm 0.04$ | $0.40 \pm 0.02$ | $0.08 \pm 0.01$ | $0.00 \pm 0.00$ | $0.02 \pm 0.00$ | $0.00 \pm 0.00$ | $0.00 \pm 0.00$ | $0.00 \pm 0.00$ |
| Dose 1  | $0.90 \pm 0.04$ | $0.51 \pm 0.03$ | $0.10 \pm 0.01$ | $0.00 \pm 0.00$ | $0.02 \pm 0.00$ | $0.00 \pm 0.00$ | $0.00 \pm 0.00$ | $0.00 \pm 0.00$ |
| Dose 2  | $3.52 \pm 0.15$ | $2.05 \pm 0.09$ | $0.58 \pm 0.03$ | $0.12 \pm 0.01$ | $0.25 \pm 0.02$ | $0.11 \pm 0.01$ | $0.01 \pm 0.00$ | $0.00 \pm 0.00$ |

Table 2: **Simulation Results: MSE** Our estimator's MSE decreases to 0 as $T$ or $n$ grows. Denoting the estimated effect $j$ (e.g., $j =$ "Blip") for replicate $r$ as $\hat{\beta}_{j,r}$, we show $\text{MSE}_j := \frac{1}{R}\sum_{r=1}^{R}(\hat{\beta}_{j,r} - \beta_j)^2$ for $R = 1000$ simulation replicates ($\pm$ SE) for a sample size, $n$, and timepoints, $T$. Values are scaled by 100 for readability (e.g., 0.01 is shown in the table as 1.0). Thus 0 indicates a value $< 1e - 4$.

|         | T = 50 | | | | T = 500 | | | |
|---------|--------|--------|--------|--------|--------|--------|--------|--------|
| Effect  | $n = 6$ | $n = 10$ | $n = 30$ | $n = 100$ | $n = 6$ | $n = 10$ | $n = 30$ | $n = 100$ |
| Blip    | $0.23 \pm 0.55$ | $0.42 \pm 0.43$ | $0.42 \pm 0.24$ | $0.10 \pm 0.13$ | $0.07 \pm 0.17$ | $0.05 \pm 0.13$ | $0.08 \pm 0.08$ | $0.03 \pm 0.04$ |
| Dissip  | $0.82 \pm 0.93$ | $1.04 \pm 0.71$ | $0.45 \pm 0.39$ | $0.26 \pm 0.22$ | $0.17 \pm 0.27$ | $0.22 \pm 0.21$ | $0.10 \pm 0.13$ | $0.06 \pm 0.07$ |
| Dose 0  | $0.28 \pm 1.48$ | $0.15 \pm 1.17$ | $0.12 \pm 0.68$ | $0.42 \pm 0.36$ | $0.28 \pm 0.48$ | $0.23 \pm 0.35$ | $0.34 \pm 0.20$ | $0.13 \pm 0.11$ |
| Dose 1  | $0.12 \pm 0.42$ | $0.05 \pm 0.32$ | $0.28 \pm 0.18$ | $0.06 \pm 0.10$ | $0.10 \pm 0.13$ | $0.08 \pm 0.10$ | $0.04 \pm 0.06$ | $0.01 \pm 0.03$ |
| Dose 2  | $0.41 \pm 0.35$ | $0.09 \pm 0.27$ | $0.12 \pm 0.15$ | $0.10 \pm 0.08$ | $0.03 \pm 0.11$ | $0.05 \pm 0.08$ | $0.03 \pm 0.05$ | $0.01 \pm 0.03$ |

Table 3: **Simulation Results: Bias** Our estimator is unbiased. Moreover, the absolute relative bias decreases to 0 as $T$ and/or $n$ grows. Denoting the estimated effect $j$ (e.g., $j =$ "Blip") for replicate $r$ as $\hat{\beta}_{j,r}$, we show Absolute Relative $\text{Bias}_j := |\frac{1}{R}\sum_{r=1}^{R}(\hat{\beta}_{j,r} - \beta_j)/\beta_j|$ for $R = 1000$ replicates ($\pm$ standard error). Values are scaled by 100 for readability (e.g., 0.01 is shown in the table as 1.0).

target pose on the next trial. The effects are significant for at least one dose value in all but two target poses. Interestingly, the effect is also significantly negative for one target pose.

**Conditional Excursion Effect**  We next estimate an availability-conditional estimand [5], to determine if existing excursion effect methods have the capacity to reveal the effects identified with our method. We estimate this in the MSM

$$\text{logit}\left(\mathbb{E}[Y_t(a_t) \mid I_t = 1, G = 1]\right) = \alpha_0 + \alpha_1 a_t. \tag{6}$$

Figure 8B shows the availability-conditional treatment effect estimates, $\widehat{\alpha}_1$ estimated in model (6). It identifies no significant effects for any target pose. The conditional estimand, often referred to as a "blip effect" (see Figure 1A for an illustration) is only defined for the effect of applying the laser on the most recent trial (i.e., a dose of 1), and thus cannot estimate dose-response profiles. In contrast, our approach can test *sequential* excursion effects (i.e., for policies with $\Delta > 1$), enabling the estimation of a dose-response profile that reveals treatment effects here. Importantly, the effect estimates $\widehat{\beta}_1$ and $\widehat{\alpha}_1$ have different interpretations because $\widehat{\beta}_1$ reflects a causal effect of a single treatment opportunity for any of the last $\Delta = 5$ trials, and $\widehat{\beta}_1$ is not interpreted as conditional on availability.

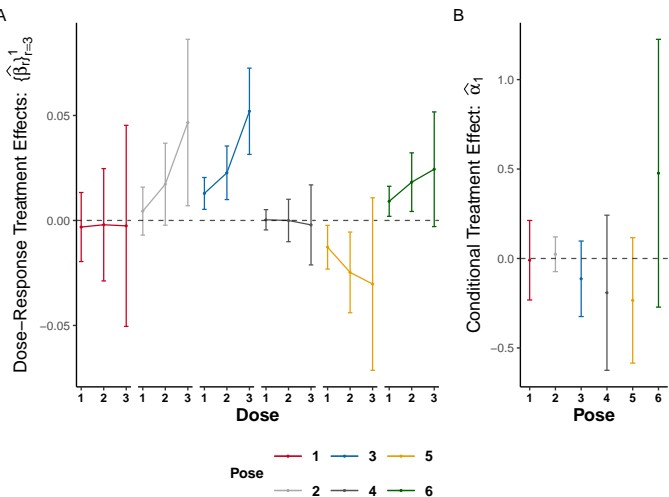

Figure 8: **Our method enables estimation of dose effects.** Plots show coefficient estimates (error bars show 95% CIs) as a function of dose. Columns and colors indicate the dose. [A] Main effects of stimulation opportunity from HR-MSM (5). [B] Availability-conditional effects of treatment estimated in MSM (6).

### E.1.1   Effect Modification by Baseline Behavior

**Marginal Effect Modification Parameter**   Next we show the capacity of our method to estimate effect modification in the form of interactions between covariates and functions of deterministic policies (i.e., treatment opportunity dose). We estimated effect modification of total target pose counts on baseline sessions, $\bar{Y}^0$ (defined in Section 4.2), because [17] estimated the treatment effect of the laser by comparing the mean change in outcome levels between treatment and baseline sessions.

Augmenting the HR-MSM in the previous section with effect modifier, $\bar{Y}^0$, we fit the HR-MSM

$$\text{logit}\left(\mathbb{E}[Y_t(\boldsymbol{d}_{\Delta,t}) \mid \bar{Y}^0, G=1,]\right) = \beta_0 + \sum_{r=1}^3 \beta_r \mathbb{1}\left(\sum_{j=t-\Delta+1}^t \sigma_j(d_j) = r\right) + \beta_4 \bar{Y}^0 + \sum_{r=1}^3 \beta_{4+r} \bar{Y}^0 \times \mathbb{1}\left(\sum_{j=t-\Delta+1}^t \sigma_j(d_j) = r\right). \tag{7}$$

Similarly, we fit an analogous model for the conditional estimand,

$$\text{logit}\left(\mathbb{E}[Y_t(a_t) \mid I_t=1, G=1, \bar{Y}^0]\right) = \alpha_0 + \alpha_1 a_t + \alpha_2 \bar{Y}^0 + \alpha_3 \bar{Y}^0 \times a_t. \tag{8}$$

We centered and scaled the effect modifier to make the coefficients easier to interpret.

Figure 9A-B shows how our method can be used to probe effect-modification. The figures illustrate that our approach estimates that for two target poses, there is a statistically significant effect modification by baseline responding at, at least, one of the doses. Specifically, animals who exhibited higher levels of responding on *baseline* sessions, exhibited a larger effect of dose. Figure 9C-D shows that the availability-conditional estimator does not identify any significant main effect of stimulation or interactions with baseline responding. These results show how our framework enables one to probe effect modification of sequential excursion effects.

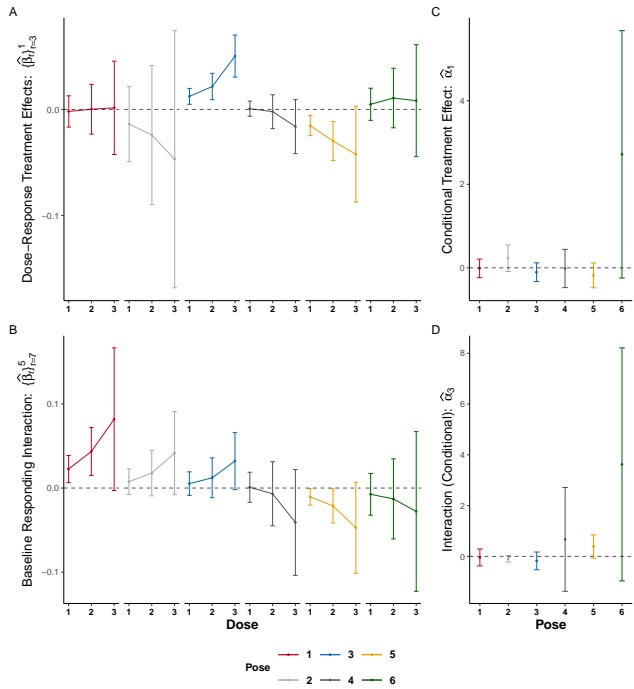

Figure 9: **Our method enables estimation of effect modification of baseline behavioral responding.** Plots show coefficient estimates (error bars show 95% CIs) as a function of dose. Columns and colors indicate pose. [A] Main effects of treatment opportunity for doses 1-3 at mean baseline responding levels with marginal HR-MSM (our approach), $\Delta = 5$ $k = 1$. [B] Interaction with baseline (pre-stimulation session) responding levels with marginal HR-MSM (our approach). [C] Main effects of stimulation on past trial at mean baseline responding levels with availability-conditional approach. [D] Interaction between stimulation and baseline responding with availability-conditional approach.

# F    Data, Pre-processing and Code Availability

We provide code to reproduce all analyses and figures on anonymous GitHub Repo: `https://anonymous.4open.science/r/causal_opto-52CD/README.md`. We downloaded the open-source dataset from [17] from `https://zenodo.org/records/7274803`. We used the open-source pre-processing code provided by the authors on Github repo `https://github.com/dattalab/dopamine-reinforces-spontaneous-behavior`. We analyzed data from all animals in the online dataset (both ChR2 and Chrimson animals). We constructed trials as described in [17]. That is, we defined trials as consecutive timepoints when the animal was classified to be in a given pose. For our HR-MSM analyses of the treatment (opto) sessions, we classified "target pose" trials only if they met the criteria of [17], which required that the hidden Markov model predictions had sufficiently high forward algorithm probabilities of the latent states. This indicator was provided in the opto session dataset provided by the authors. The baseline session data did not, however, include this indicator since no optogenetic stimulation was applied. Thus when recreating the "standard" between-group (macro longitudinal) analyses that compared baseline and opto session data, we did not classify target pose trials based on whether it met this criteria: we classified the pose based on the most likely latent state prediction but did not require the forward algorithm probabilities met the threshold set by the authors (for either baseline or opto sessions to be consistent). We corresponded closely with the authors to ensure we pre-processed the data correctly.

There was a small percentage of trials that the authors described eliminating because they were deemed too short. We did not eliminate these trials because this created inconsistencies in the pattern of trials: it allowed two consecutive trials to be of the same trial type which broke with the pattern in the remainder of the dataset. This was a very small percentage of the dataset. We compared results with and without this criteria and the decision appeared to have negligible effects on analysis results.

Finally, to the best of our understanding, the original authors' hypothesis tests were conducted on further processed version of the data that first calculated the number of target pose occurrences in each 30 second bin of the experiment (period). From our understanding, this was done to provide a smoothed time-series of outcome frequency across the course of the opto sessions. We conducted similar analyses to make sure our pre-processing yielded comparable results, but we did not use these pre-processing steps in our HR-MSM analyses or replication of the "standard" between-group (macro longitudinal) analyses as it appeared to "double-count" target pose occurrences that began before the end of one 30 second bin and ended after the start of the subsequent bin.

Finally, as described in [17], the experiment included two 30 minute replicates of both opto and baseline sessions. We constructed trials on each replicate separately (to account for the discontinuity in time between replicates) and then pooled the replicate datasets together to be consistent with the analysis procedures in [17]. We accounted for the longitudinal structure by using sandwich variance estimators in all of our hypothesis tests.

### F.1 Replication of Original Author Analysis

Finally, Figure 3D shows that the "standard" macro longitudinal effects exhibit no significant changes, emphasizing how estimands that marginalize over the stochastic dynamic (closed-loop) policies can obscure effects. We visualize the within-subject differences in target pose counts between treatment (opto) and baseline sessions in Figure 10: $\sum_{t=1}^{T} Y_t - \sum_{t=1}^{T_0} Y_t^0$, where $Y_t, Y_t^0 \in \{0, 1\}$ are indicators that the animal exhibited the target pose on trial $t$ of the treatment and baseline sessions, respectively; $T, T_0 \in \mathbb{Z}$ denote the total number of trials in treatment and baseline sessions, respectively. Because of the trial definition, the total number of trials usually differed within-subject between treatment and baseline sessions (i.e., $T \neq T_0$), but we found comparable analyses of $\frac{1}{T} \sum_{t=1}^{T} Y_t$ and $\frac{1}{T_0} \sum_{t=1}^{T_0} Y_t^0$ yielded similar results to analyses of total counts $\sum_{t=1}^{T} Y_t - \sum_{t=1}^{T_0} Y_t^0$. For that reason, we present results in terms of total outcome counts to be consistent with the analyses presented in [17]. We showed these results to the authors of [17] and they confirmed that these analyses aligned with theirs.

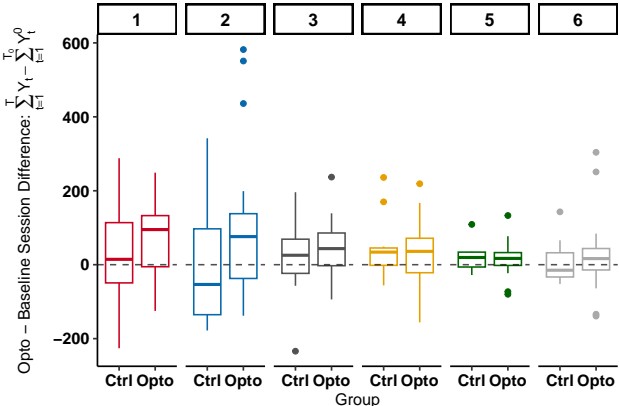

Figure 10: **Difference between target pose counts within-subject between baseline and treatment (opto) sessions.** Each point in the boxplot shows $\sum_{t=1}^{T} Y_t - \sum_{t=1}^{T_0} Y_t^0$, where $Y_t, Y_t^0 \in \{0, 1\}$ are indicators that the animal exhibited the target pose on trial $t$ of the treatment and baseline sessions, respectively. $T, T_0 \in \mathbb{Z}$ were the total number of trials in treatment and baseline sessions, respectively. Columns and colors indicate target pose. *Ctrl* and *Opto* indicates control and treatment group subjects, respectively.

