# OpenReview forum: "Causal Inference in the Closed-Loop: Marginal Structural Models for Sequential Excursion Effects"
_NeurIPS.cc/2024/Conference — NeurIPS 2024 poster_

### Official Review · Reviewer_manL · 2024-07-10

**Soundness:** 4
**Presentation:** 3
**Contribution:** 3
**Rating:** 7
**Confidence:** 2

**Summary:**

The paper proposes novel causal inference framework for closed-loop optogenetics behavioral studies, develops causal inference estimation method for sequential excursion effects that capture local causal contrasts within the same treatment group. The proposed method is robust to accommodate both positivity violations and longer treatment sequences. The paper provided a practical computational implementation and analyze the method with theoretical guarantees. More interestingly, it empirically corroborates their studies with real-applications.

**Strengths:**

The utility and implications of counterfactual causal effect under dynamic treatment policies are well-described by authors. I found it especially interesting that the authors performs empirical validation through applying their method on existing Nature paper and found their computational result is corroborated with the original authors of the existing paper. This shows a thorough post-hoc validation of the proposed method.

**Weaknesses:**

The largest weakness I see here is both experimental results uses linear or generalized linear model for $m$. I see the advantages of well-developed generalized linear model in experimental sciences for good interpretation and statistical guarantees, though would be curious to find out how the authors see the method would work beyond linear models.

**Questions:**

- Proposition 3.4 $Z_i$ is introduced but the proposition only uses $Z$ throughout. would be good to specify the relationship between $Z_i$ and $Z$.

**Limitations:**

Yes.

---

> ### Author Rebuttal · Authors · 2024-08-06
>
> We thank the reviewer for their positive feedback on our work. The reviewer raises a very interesting methodological/modeling question, and identifies a notational issue, both of which we address below.
>
> 1. “The largest weakness I see here is both experimental results uses linear or generalized linear model for $m$. I see the advantages of well-developed generalized linear model in experimental sciences for good interpretation and statistical guarantees, though would be curious to find out how the authors see the method would work beyond linear models.”
>
> - The projection parameter specification, as laid out in Section 3, is entirely generic, and remains well-specified even for more complicated models $m$ (e.g., random forests, kernel-based methods, neural networks). However, as you rightly point out, we adopt linear or generalized linear models (GLMs) as our working models in both simulations and the applied study. For the questions of scientific interest in the applied optogenetics study, we felt that the target causal effects (e.g., dose response effects and effect dissipation, pooled across time) could be well approximated using GLMs.
>
> - The theory applies for a large class of working mean models $m$, and does not rely on any distributional/likelihood-based assumptions. Beyond their familiarity to most readers, the technical reason for opting for GLMs in our applied examples is—as you again correctly identify—for statistical guarantees (e.g., convergence properties and inferential tools). In particular, Theorem 3.5 stipulates that $m$ must be “Donsker” in $\beta$: GLMs automatically satisfy this technical requirement (see, e.g., Examples 19.7 and 19.8 in van der Vaart (2000)). Whether a model class is “Donsker” is a non-trivial query regarding its “complexity” or “size”, but can be investigated using machinery and results from empirical process theory (van der Vaart & Wellner (1996)). The situation is indeed more complicated for more flexible models, and this Donsker requirement is not always satisfied. However, we note that it *is* satisfied for some formulations of random forests (see, e.g., Theorem 3.2 in Scornet (2016)) and kernel estimators (see, e.g., Theorems 3.7 and 3.13 in Beutner & Zähle (2023)). In our revised Discussion section, we will discuss this issue in more detail, and note that future research will focus on applying random forests and kernel estimators (when they satisfy the Donsker requirement), and the more challenging problem of studying the statistical properties of flexible model specifications when the Donsker requirement is not satisfied.
>
> 2. “Proposition 3.4 $Z_i$ is introduced but the proposition only uses $Z$ throughout. would be good to specify the relationship between $Z_i$ and $Z$.”
>
> - We thank the reviewer for noticing this notational issue. As we mention in the beginning of Section 2, we adopt a common statistical shorthand in that “we often suppress [subject-specific] indices to reduce notational burden”. In this case, $Z$ represents a generic observation (i.e., from an arbitrary subject), whereas $Z_i$ represents that same data, but specifically from subject $i$. In the revision, we will specifically reiterate in Proposition 3.4 that we omit the subject-specific index for clarity.
>
> **References**:
> - van der Vaart AW, Wellner JA. Weak convergence. Springer New York; 1996.
> - van der Vaart AW. Asymptotic statistics. Cambridge University Press; 2000 Jun 19.
> - Scornet E. On the asymptotics of random forests. Journal of Multivariate Analysis. 2016 Apr 1;146:72-83.
> - Beutner E, Zähle H. Donsker results for the empirical process indexed by functions of locally bounded variation and applications to the smoothed empirical process. Bernoulli. 2023 Feb;29(1):205-28.

---

> > ### Comment · Reviewer_manL · 2024-08-10
> >
> > Thank you for the comprehensive rebuttal.

---

### Official Review · Reviewer_z4yk · 2024-07-11

**Soundness:** 3
**Presentation:** 2
**Contribution:** 2
**Rating:** 5
**Confidence:** 2

**Summary:**

This paper introduces a causal inference framework for analyzing close-loop optogenetics experiments. The authors propose using HR-MSMs to estimate the sequential excursion effects (causal effects of specific neural stimulation sequences on behavioral outcomes). Specifically, they develop an IPW estimator with a scalable implementation and provide the theoretical guarantee of consistency and asymptotic normality under mild assumptions. The proposed framework extends the existing approaches to handle longer treatment sequences and addresses their limitations such as positivity violations. Through both simulation studies and an application to a real-world optogenetics study, the authors show that the proposed framework can reveal longitudinal causal effects that are often obscured by standard optogenetics analysis methods due to "treatment-confounder" feedback in closed-loop designs.

**Strengths:**

1. This paper proposes the first formal causal inference framework based on HR-MSMs for closed-loop optogenetics behavioral studies, addressing the challenges (e.g., positivity violations and "treatment-confounder" feedback) in the standard methods.

2. The authors prove the consistency and asymptotic normality of the proposed IPW estimator, under mild assumptions.

3. The proposed framework is evaluated on both simulated and real optogenetics studies, and demonstrates its effectiveness in analyzing more complex causal effects in closed-loop designs than standard analyses.

**Weaknesses:**

1. The focus of this paper centers on an application study that refines causal effect estimation within a specialized causal graph, akin to Figure 1E, rather than introducing broad methodological advancements in causal inference. The authors adapt Marginal Structural Models (MSMs) to dynamic treatment regimes and introduce History-Restricted MSMs (HR-MSMs) for estimating sequential excursion effects. This setup closely relates to estimating causal effects in temporal sequences. It is unclear what is the difference/gap between the proposed framework and the existing temporal causal inference methods. I wonder if the authors could discuss this in more detail. For instance, can any temporal causal inference framework estimate sequential excursion effects accurately? What are the challenges in performing causal inference the closed-loop optogenetics behavioral studies compared to more general (spatial-)temporal studies?

2.  I think it would be better if the authors explicitly introduce the problem settings in Section 2 rather than mention it together with relevant literature. The current relevant literature part involves many technical details, which makes it slightly difficult for readers to understand the gap between the proposed method and existing works.

3. The authors should consider adding more baselines in experiments. In this paper, the authors did not use any baselines, and thus it is unclear whether the proposed framework outperforms existing methods.

4. The settings and evaluation metrics of the experiments need to be revised.

a. I think one of the advantages of the proposed method is to handle more complex local causal effects or longitudinal effects, whereas the standard methods only estimate macro effects. Intermediate causal effects can be canceled out in some cases, so only estimating macro effects can be biased and inaccurate. However, the authors did not conduct any simulation studies to demonstrate the advantage of the proposed method.

b. The current presentation of experimental results primarily relies on descriptive analysis, and I think the authors should use some quantitative metrics to evaluate the performance.

**Questions:**

1. Please see the questions in the Weaknesses part.

2. I wonder how is $\Delta$ determined and How would the value of $\Delta$ affects the sequential excursion effects?

**Limitations:**

The authors adequately addressed the limitations of their work.

---

> ### Author Rebuttal · Authors · 2024-08-06
>
> We thank the reviewer for their helpful questions, comments and feedback on our work. We believe that the proposed revisions and clarifications in line with the responses below will improve the strength of the paper.
>
> 1. “The focus of this paper centers on an application study that refines causal effect estimation within a specialized causal graph, akin to Figure 1E, rather than introducing broad methodological advancements in causal inference.”
>
> - We used the causal graph in Figure 1E to provide an illustration, which may have suggested limited applicability. To clarify, the proposed methods are applicable in **all sequentially randomized settings**, going far beyond the optogenetics studies that motivated our work. In particular, treatment at each time point may depend on the *complete* history of observed variables, $H_{t}$ (as opposed to just $X_t$ as in the illustrative causal graph). In the revision, we will emphasize that this is an illustrative causal graph, but elaborate on the generality of the methodology to causal effect estimation within a far more general set of causal graphs. The methodological advancement is the introduction of specific estimands that are valid in the presence of positivity violations, paired with powerful statistical approaches. Such positivity violations occur regularly across many application areas beyond neuroscience, for instance in mobile health studies where active treatment may be systematically withheld for ethical reasons (e.g., see discussion of treatment “availability” in Section 4 in Boruvka et al. (2018; JASA)).
>
> 2. “[...] This setup closely relates to estimating causal effects in temporal sequences. It is unclear what is the difference/gap between the proposed framework and the existing temporal causal inference methods. I wonder if the authors could discuss this in more detail. [...]”
>
> - Please see global “Author Rebuttal” for a detailed response: the major challenges in this case are the large number of timepoints (for which traditional MSMs are unstable) and the inherent positivity violations in the design (where no existing methods can handle treatment sequences longer than length one). Further, existing machine learning-based causal methodology (RMSNs, CRNs, CTs) rely on positivity, and do not provide statistical guarantees or inferential tools (e.g., confidence intervals).
>
> 3. “I think it would be better if the authors explicitly introduce the problem settings in Section 2 rather than mention it together with relevant literature…”
>
> - We thank the reviewer for this suggestion and agree with their point. In the revision, we will split Section 2 into two sections. The first will be a less technical summary of existing methods, highlighting the gaps. The second will then explicitly introduce the problem and notation.
>
> 4. “The authors should consider adding more baselines in experiments. In this paper, the authors did not use any baselines, and thus it is unclear whether the proposed framework outperforms existing methods.”
>
> - Please see global “Author Rebuttal” for a detailed response. Briefly, our paper introduces new target causal estimands. Consequently, no existing estimators could be used to compare against our approach. That said, we will add a simulation in the appendix (see Figure 1 in attached 1-page PDF) to show how the proposed methods uncover effects when standard “macro” approaches yield null results (see Point 6 below).
>
> 5. “The settings and evaluation metrics of the experiments need to be revised.”
>
> - We thank the reviewer for their suggestion; we believe this criticism reflects our choice to present our findings graphically as opposed to numerically (e.g., in tables). In the revision, we will provide the numerical bias and coverage results in a table, and display mean-squared error (MSE) of the proposed estimator across the simulation settings (see Table 1 in attached 1-page PDF).
>
> 6. “[...] so only estimating macro effects can be biased and inaccurate. However, the authors did not conduct any simulation studies to demonstrate the advantage of the proposed method. ”
>
> - In Appendix 2, we show mathematically that macro effects may be misleading. Specifically, we demonstrate that, even if the magnitude of local causal effects are large, the macro effects can be exactly zero. In the revision, we will include a simulation that demonstrates this more concretely: macro approaches will show null results, whereas the proposed methods will uncover more nuanced non-null effects (see Figure 1 in attached 1-page PDF for preliminary results).
>
> 7. “The current presentation of experimental results primarily relies on descriptive analysis, and I think the authors should use some quantitative metrics to evaluate the performance.”
>
> - See response to Point 5 above: we will provide explicit numerical tables (in addition to our figures) to demonstrate bias, MSE, and coverage properties (see Tables 1-3 in attached 1-page PDF).
>
> 8. “I wonder how is $\Delta$ determined and How would the value of $\Delta$ affects the sequential excursion effects?”
>
> - The number of intervention timepoints in the counterfactual outcome of interest ($\Delta$) determines the nature of the effects being estimated. However, 2-3 intervention timepoints are often sufficient to capture a rich collection of sequential excursion effects. We found that, in our application, conclusions were stable across a range of $\Delta$ values (e.g., in Section 4.2, we mention that “[w]e fit comparable models for sequences as long as $\Delta=7$ and found results were similar across $\Delta$ values.”). In practice, the value of $\Delta$ may be constrained, as the variance of parameter estimates can grow if $\Delta$ is set too large (likely for the same reason that the parameters of non-history-restricted MSMs grows prohibitively large as the number of timepoints increases). In our revision, we will emphasize that $\Delta$ should be chosen on the basis of subject matter knowledge.

---

> ### Comment · Reviewer_z4yk · 2024-08-13
> **Official Comment by Reviewer z4yk**
>
> Thank you for the detailed response and the additional numerical experimental results. I have raised my score by 1.

---

### Official Review · Reviewer_pQ5n · 2024-07-13

**Soundness:** 3
**Presentation:** 3
**Contribution:** 3
**Rating:** 4
**Confidence:** 2

**Summary:**

This work investigate the causal effect estiamtion in the Optogenetics, which is interestimg in causality in science. The paper proposed a nonparametric causal inference framework for analyzing “closed-loop” designs in sequential setting and  extends “excursion effect” methods to enable estimation of causal contrasts for treatment sequences. The writing is good, and the toptic is important for the causality and biologic community.

**Strengths:**

1. The investigated problem is important.

2. The presentation and writing is good.

**Weaknesses:**

1. Some claims are not clear, which makes the paper unreadable, especially for those who are not familiar on biologic and optogenetics

2. The experimental evaluation is weak.

**Questions:**

1.I am kind of confused for the definition and notation of treatment, what's the difference of G and A? I thought they are the same but the description in intro confuses me. optogenetic manipulation is not about the laser and no-laser.

2. The experimental evaluation is weak. Indeed there are only one baseline method, i.e., HC to compare with the proposed estimator, which is is too limited, and it is not convincing that the proposed method is effective.

3.There are many sequential models (e.g., Causal Transformer, CRN, RMSN ,etc) which can handle this kind of time-dependent causal inference task.

---

> ### Author Rebuttal · Authors · 2024-08-06
>
> We thank the reviewer for their feedback and questions. Most of the major concerns are, we believe, a result of misunderstandings of the goals and claims of our work. We hope that our revisions will make them clearer, and address the questions below:
>
> 1. "Some claims are not clear, which makes the paper unreadable, especially for those who are not familiar on biologic and optogenetics."
>
> - The reviewers unanimously felt that a strength of our work is the importance of the methods for applications in neuroscience, thus we feel we should maintain some domain-specific language. We make a significant effort to provide the necessary background to motivate our methods. Moreover, the problem setup and counterfactual notation we use follow the conventions of the sequentially randomized experiments causal inference literature (e.g., MSMs, HR-MSMs) as well as much of the causal machine learning research (e.g., RMSNs, CRNs, CTs). Thus, no domain-specific knowledge should be necessary to understand the key methodological and theoretical development (Sections 2, 3, 4.1). Our contributions to the field of counterfactual-based causal inference are general and extend well-established statistical theory.
>
> 2. "I am kind of confused for the definition and notation of treatment, what's the difference of G and A? I thought they are the same but the description in intro confuses me. optogenetic manipulation is not about the laser and no-laser."
>
> - We are sympathetic that the biology may be unfamiliar to some readers and, in our revision, we will further emphasize the distinction between group ($G$) and laser ($A$). The group, $G$, and laser stimulation, $A$, are indeed very different. Optogenetics studies often include randomly assigned groups of “light-sensitive” animals ($G = 1$), and light-insensitive negative controls ($G = 0$). Animals in the $G = 1$ group express a protein that causes their neurons to be activated in response to laser stimulation, whereas negative control animals ($G = 0$) do not express this protein, and thus do not exhibit this response to the laser. Animals in *both* groups can receive the laser treatment ($A_t = 1$), or not ($A_t = 0$), on each trial, $t$, but the application of the laser will not influence the neuronal activity in the control group ($G=0$) . This is a fundamental distinction, which we make a significant effort to explain in the paper. Indeed, on line 32 we state: “Experiments often include both treatment ($G=1$) and negative-control ($G=0$) groups, with animals assigned randomly to each. While the laser is often applied on a random subset of trials in both groups, only treatment animals express the protein that enables the laser to trigger the target neural response. The control group thus controls for “off-target” effects such as the laser heating the brain…”
>
> 3. "The experimental evaluation is weak. Indeed there are only one baseline method, i.e., HC to compare with the proposed estimator,"
>
> - See global “Author Rebuttal for a detailed response. A primary contribution of our work is the introduction of **new target estimands** (i.e., they allow us to ask new causal questions that previous methods cannot): there are *no existing estimators* against which we can compare our proposed methods.
>
> - We believe that the reviewer’s description of our empirical evaluation as “weak” may be a result of misunderstanding. For example, “HC” is *not* the “only one baseline method.” Rather “HC” is a family of variance estimators that can be used to estimate uncertainty for our point estimators, i.e., they can be used as components of our proposed framework. We describe in the Results Section 4.1 that: “the sample size-adjusted HC3-based CIs achieve 95% coverage. All CIs achieve 95% coverage when n is large, showing that we can conduct valid inference in all settings.” Moreover, in the Figure 2 caption, where the “HC” results are presented, we describe that the results show “[t]he coverage of 95% CIs constructed using one of three established robust variance estimators and our robust variance estimator. The nominal coverage is reached for either large $n$ or large $T$ for all estimators.”
>
> - As we emphasize in the global response, we evaluated our methods across a wide range of simulation settings (see Tables 1-3 in the attached 1-page PDF with updated numerical results), and a very thorough application for which the results were confirmed and commended by the original neuroscientists who authored the Nature paper.
>
> 4. "There are many sequential models (e.g., Causal Transformer, CRN, RMSN ,etc) which can handle this kind of time-dependent causal inference task."
>
> - As we detail in the global response, we omitted these methods because they are not applicable in our setting. In our revision, we will expand our literature review to include them, and explain the reasons why they are not suitable for this problem:
>
> “While RMSNs (Lim et al., 2018), CRNs (Bica et al., 2020) and CTs (Melnychuk et al., 2022) can be used to estimate causal effects over time, these methods have important limitations: (a) these methods all target the effects of **static** (i.e., fixed) treatment sequences and depend *heavily* a positivity assumption, and thus cannot be applied in closed-loop designs; (b) the target estimands for these methods are *always* conditional on complete history of covariates/outcomes before treatment initiation, $H_{t - \Delta + 1}$, whereas our estimands may depend on all history variables if desired, but may be averaged over some or all of these variables (often yielding greater statistical power) depending on the user’s preference; and (c) unlike for our methods, there are no rigorous statistical guarantees (e.g., consistency, asymptotic normality) that have been proven for RMSNs, CRNs or CTs, and correspondingly, no inferential tools (e.g., confidence intervals) developed for uncertainty quantification.”

---

### Author Rebuttal · Authors · 2024-08-06

We would like to thank the reviewers for their thoughtful feedback, and were pleased to see agreement on the following positive points:

**Importance & Novelty**:
- z4yk: “This paper proposes the first formal causal inference framework…for closed-loop optogenetics behavioral studies addressing the challenges (e.g., positivity violations and "treatment-confounder" feedback) in the standard methods.”
- manL: “The paper proposes novel causal inference framework for closed-loop optogenetics behavioral studies.”
- pQ5n: “Causal effect estimation in [o]ptogenetics…is interesting in causality in science...and the topic is important for the causality and biologic community.”

**Methodological Contributions**:
- z4yk: “The ... framework extends the existing approaches to handle longer treatment sequences and addresses their limitations...and [provides a] scalable implementation and… theoretical guarantee of consistency and asymptotic normality under mild assumptions.”
- manL: “The paper develops causal inference estimation method for sequential excursion effects that capture local causal contrasts within the same treatment group. The ... method is robust to ... positivity violations and longer treatment sequences.The paper provided a practical computational implementation and analyze the method with theoretical guarantees.“

**Experimental Evaluation**:
- manL:  “... the authors perform empirical validation through applying their method on existing Nature paper and found their computational result is corroborated with the original authors of the existing paper. This shows a thorough post-hoc validation of the proposed method.”
- z4yk: “The proposed framework is evaluated on both simulated and real optogenetics studies, and demonstrates its effectiveness in analyzing more complex causal effects in closed-loop designs than standard analyses.”

**Presentation & Writing**:
- manL: “The utility and implications of counterfactual causal effect under dynamic treatment policies are well-described.”
- pQ5n: “The presentation and writing is good.”


There were also concerns raised by more than one reviewer, which we address below, leaving other points for individual reviewer responses.

**Experimental Evaluation**:

- z4yk: “The authors did not use any baselines, and thus it is unclear whether the proposed framework outperforms existing methods.”
- pQ5n: “The experimental evaluation is weak. ... there are only one baseline method, i.e., HC to compare with the proposed estimator, which is is too limited.”

**Response**

A key contribution of our work is the introduction of new target estimands. The motivation for this work is precisely the fact that no existing estimands or estimators were suitable for closed-loop designs, so there is no appropriate method to compare against. There are also no consensus benchmark datasets to analyze.

Given this, we respectfully disagree that the empirical evaluation of the methodology is “weak”. We followed two evaluation strategies:
1. application to realistic simulated data, with known ground truth
2. application to real data, then validating the results with domain experts

For 1., we simulate data across six sample sizes and three total timepoint scenarios (Apdx. Fig. 5), which cover all optogenetics applications we have seen in the literature. We show that our estimators are unbiased and that nominal coverage (e.g., 95%) can be achieved in nearly all scenarios, thus verifying our theoretical statistical results.
For 2., we analyzed a landmark optogenetics study, and have replicated and expanded the authors’ findings.

The “HC” standard error estimators are **not** competitors—these provide uncertainty quantification for our proposed point estimator. Beyond the large sample variance we derived, these alternatives can improve coverage in small sample sizes. To avoid the confusion from labeling the variance estimator $\hat{V}$ from Section 3 as “Ours”, we will relabel it as “Large Sample” in the revision.

**Distinction from Existing Methods**

- z4yk: “It is unclear what is the difference/gap between the proposed framework and the existing temporal causal inference methods.”
- pQ5n: “There are many sequential models (e.g., Causal Transformer, CRN, RMSN ,etc) which can handle this kind of time-dependent causal inference task.”

**Response**

While our methods estimate causal effects in sequential settings, standard approaches like traditional MSMs often do not perform well when there are a large number of timepoints (e.g., as in many optogenetics studies). In our revision, we will emphasize that MSM parameter variances grow prohibitively large as the number of timepoints rise, an issue that our method does not suffer from for reasonable values of $\Delta$. We will also describe how existing history-restricted MSMs rely on a positivity assumption which is violated in closed-loop designs, and/or cannot accommodate longer treatment sequences.

RMSNs, CRNs and CTs can be used to estimate causal effects over time, but have limitations that preclude their use in our setting:

- no statistical guarantees (e.g., asymptotic normality) or inferential tools (e.g., confidence intervals)
- these methods target the effects of **static** (i.e., fixed) treatment sequences and depend heavily on a positivity assumption, and thus cannot be applied in closed-loop designs
- the target estimands are *always* conditional on complete history before treatment initiation, $H_{t - \Delta + 1}$, whereas our estimands may depend on all history or averaged over some/all of these variables (often improving statistical power), if desired

To address this, we will describe these methods and their lack of suitability for closed-loop optogenetic studies in our revision.

**1-page PDF**

We include:
- New simulation results showing how treatment-confounder feedback can obscure effects
- Updated bias results from simulations, showing unbiasedness
- Tables of simulation results: MSE, bias & coverage

---

### Decision · Program_Chairs · 2024-09-25

**Decision:**

Accept (poster)

**Comment:**

This submission carefully develops novel methodology for a concrete application area. I consider the fact that it is specifically tailored to what could be viewed as a "narrow application area" a strength rather than a weakness. This led the authors to develop a new, suitable target estimand in their setting taking into account the real-world nuances (e.g., positivity violations). Both the theoretical underpinnings as well as the experimental validation are overall convincing to me (and most reviewers). Given that it is difficult to squeeze a detailed domain/problem description for non-experts into the limited space, the paper is overall well written and structured and was mostly considered fairly easy to follow and read. Ultimately, I believe this paper is going to be relevant and of interest to a substantial fraction of the NeurIPS community and the ideas developed here may be of interest also outside the specific application area. I encourage the authors to work in some of the elucidating clarifications from the rebuttal to further strengthen the paper and make it as widely accessible as possible.